# Dynamic Sparse Training of Diagonally Sparse Networks

**Abhishek Tyagi** [1]  **Arjun Iyer** [2]  **William H Renninger** [2]  **Christopher Kanan** [1]  **Yuhao Zhu** [1]

## Abstract

Recent advances in Dynamic Sparse Training (DST) have pushed the frontier of sparse neural network training in structured and unstructured contexts, matching dense-model performance while drastically reducing parameter counts to facilitate model scaling. However, unstructured sparsity often fails to translate into practical speedups on modern hardware. To address this shortcoming, we propose DynaDiag, a novel structured sparse-to-sparse DST method that performs at par with unstructured sparsity. Dyna-Diag enforces a diagonal sparsity pattern throughout training and preserves sparse computation in forward and backward passes. We further leverage the diagonal structure to accelerate computation via a custom CUDA kernel, rendering the method hardware-friendly. Empirical evaluations on diverse neural architectures demonstrate that our method maintains accuracy on par with unstructured counterparts while benefiting from tangible computational gains. Notably, with 90% sparse linear layers in ViTs, we observe up to a 3.13x speedup in online inference without sacrificing model performance and a 1.59x speedup in training on a GPU compared to equivalent unstructured layers. Our source code is available at https://github.com/horizon-research/DynaDiag/.

## 1. Introduction

Over the years, deep neural networks (DNNs) have grown, and their performance on complex tasks has increased to and beyond human-level performance (Sparkes, 2023; Lykiardopoulou, 2023). However, the cost of training and inference for such large DNNs has skyrocketed (Cottier, 2023).

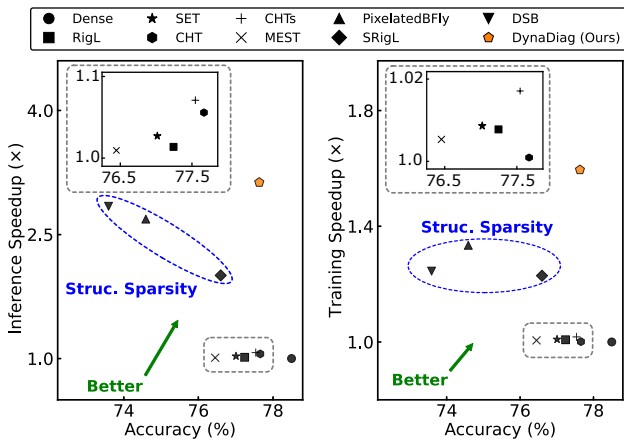

Figure 1: Comparing the inference (left) and training speedups (right) (calculated using wall-clock time) of sparse training methods and the Top-1 classification accuracy (x-axis) for a ViT-Base model at 90% sparsity running ImageNet-1K. DynaDiag, being closest to the top right corner, demonstrates superior accuracy and speedup compared to structured and unstructured sparse training approaches.

One way to reduce the execution cost of these networks and still perform at par with their dense counterparts (Frankle & Carbin, 2018; Blalock et al., 2020; Mostafa & Wang, 2019) is to compress them by removing unnecessary weights using methods such as pruning (Molchanov et al., 2016; Tanaka et al., 2020), and sparse training (Jaiswal et al., 2022; Zhang et al., 2023b).

Weights are typically pruned either randomly (unstructured sparsity (Evci et al., 2020; Han et al., 2015)) or in patterns (structured sparsity (Liu et al., 2019)). Unstructured sparsity achieves high sparsity ratios with minimal performance loss but lacks hardware acceleration. Structured sparsity, while hardware-friendly, has yet to match the performance of unstructured approaches.

Current structured sparsity methods face two key issues. **First**, they often use dense backpropagation (Lasby et al., 2023), resulting in negligible training speedup. Even when sparse gradients are computed, transposing weight matrices in backpropagation breaks hardware-friendly patterns, hindering acceleration (Hubara et al., 2021). **Second**, these methods struggle with high sparsity, suffering significant

[1]Department of Computer Science, University of Rochester, Rochester, NY, USA [2]The Institute of Optics, University of Rochester, Rochester, NY, USA. Correspondence to: Abhishek Tyagi <atyagi2@ur.rochester.edu>.

*Proceedings of the 42nd International Conference on Machine Learning*, Vancouver, Canada. PMLR 267, 2025. Copyright 2025 by the author(s).

performance drops—as we show later.

To address these limitations, we introduce a novel sparse pattern inspired by small-world networks (Watts & Strogatz, 1998; Telesford et al., 2011)—diagonal sparsity—that retains its structure during backpropagation, enabling efficient training acceleration. We propose DynaDiag, a fully differentiable training method to learn diagonal sparsity by dynamically selecting and updating the most critical diagonals during training. Our approach outperforms existing structured sparsity methods across tasks and sparsity levels, achieving high performance and efficiency, even at extreme sparsities. Experiments show that diagonal sparsity consistently surpasses structured sparse architectures in vision and language tasks while maintaining computational benefits.

Fig. 1 presents a comparison of DynaDiag with existing Dynamic Sparse Training (DST) methods for both structured and unstructured sparsity. DynaDiag achieves the highest accuracy among the structured DST methods. Moreover, DynaDiag significantly reduces the inference and training wall-clock times on a GPU.

The following are the major contributions of our work:

1. We introduce a diagonal sparsity pattern inspired by small-world networks that preserve its structure under transposition and are efficiently accelerated on GPUs.

2. We propose DynaDiag, a differentiable TopK-based Sparse-To-Sparse training algorithm to obtain diagonally-sparse neural networks.

3. We conduct extensive empirical evaluations of DynaDiag on computer vision and natural language tasks, demonstrating that it outperforms all prior structured sparsity methods under the same sparsity budget. Additionally, DynaDiag achieves competitive performance, showing no statistically significant difference compared to unstructured sparsity techniques like RigL (Evci et al., 2020).

4. We introduce a method to convert diagonally sparse matrices to Block CSR (BCSR) format to enable speedups in both inference and training.

## 2. Related Work

### 2.1. Sparsity in Neural Networks

The main idea behind sparsity in neural networks is to remove the weights or activations that have minimal contribution to the model's overall performance. The most common way of doing this is to pre-train a dense network and remove unimportant weights using heuristics such as weight magnitude (Han et al., 2015) or gradients (Molchanov et al., 2017; 2019; Lee et al., 2018; Wang et al., 2020) and then fine-tune the model to maintain the accuracy. This approach

is commonly known as *Pruning* and has been used to compress CNNs (Cai et al., 2022; Lin et al., 2019), ViTs (Yang et al., 2023b; Yu et al., 2022) and LLMs (Lu et al., 2024). Lottery Ticket Hypothesis (LTH) (Frankle & Carbin, 2018) states that within a large, randomly initialized neural network, there exists a smaller, sparse subnetwork (a "winning ticket") that, when trained in isolation from the original initialization, can match or exceed the performance of the full, dense model and potentially be much more efficient to train from scratch.

### 2.2. Sparse Training Methods

Sparse training aims to train a sparse neural network from scratch. It can broadly be classified into either Static Sparse Training (SST) or Dynamic Sparse Training (DST).

In SST, the positions of the non-zeros in each weight matrix are fixed at the start of the training and are maintained the same throughout. Training then optimizes the values of the non-zeros for loss minimization. Pixelated Butterfly (Dao et al., 2021) uses butterfly factorization to fix the mask at initialization. However, SST is prone to a higher loss than DST as DST can escape the local minimum (Evci et al., 2020), especially at high sparsities.

In DST, the positions of the non-zeros are updated dynamically during training. The usual way to do DST is by starting with sparse weight matrices before repeatedly training the network for a few iterations, then removing the weights that are of least importance based on magnitude (Mocanu et al., 2018; Jayakumar et al., 2020) or gradients (Evci et al., 2020; Chen et al., 2022), and then growing another set of connections which will be trained in the next iterations.

SET (Mocanu et al., 2018) is one of the earliest DST works that introduced a prune-and-regrow strategy for DST, where during the prune phase, weights are pruned based on their magnitude and are regrown randomly. MEST (Yuan et al., 2021) regrows weights randomly and uses a combination of weight magnitude and gradient magnitude of the existing weights to prune them. RigL (Evci et al., 2020), on the other hand, prunes weights based on their magnitudes and are regrown based on the gradients of missing links (zero weights), which makes the backward pass dense and unable to take advantage of the sparsity in the network. Addressing this limitation of RigL, Zhang et al. (2024) proposes CHT and CHTs (Zhang et al., 2025) methods where a gradient-free (and based on the network topology) approach is used during the regrow phase, which makes their method scalable and achieves state-of-the-art performance at broad range of sparsities.

Top-KAST (Jayakumar et al., 2020) is closest to our work and uses a TopK function to pick the most useful weights in both the forward and backward pass. However, unlike Dyna-

Diag, both methods result in an unstructured distribution of nonzero weights and, hence, do not yield speedups on GPUs. SRigL (Lasby et al., 2023) addresses the above limitations

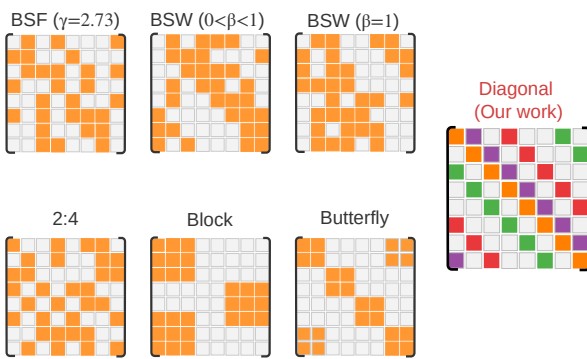

Figure 2: Overview of different sparsity patterns from the literature used in sparse training methods. Bipartite Scale-Free (BSF) and behavior of Bipartite Small-World (BSW) networks with varying $\beta$ is explained in Apdx. I

of RigL by dynamically identifying weight matrices that abide by N:M sparsity pattern, which can be accelerated on GPUs (by transforming N:M to 2:4 pattern supported by NVIDIA GPUs (Mishra et al., 2021; Hu et al., 2024)). The authors show that SRigL retains the same accuracy as RigL for ViTs and CNNs, though the method lacks evaluations on large language models (LLMs). Moreover, while they report inference speedups in the final trained models, the training process does not currently benefit from this sparsity. DSB (Jiang et al., 2022) uses block sparsity to accelerate the training and inference while dynamically looking for the optimal placement of non-zero blocks. However, as we show in Sec. 4, block sparsity loses out significantly at high sparsities like the pixelated butterfly method.

The various structured and unstructured sparsity patterns discussed are shown in Fig. 2. We will now describe the training of diagonally sparse neural networks.

## 3. Training Diagonally Sparse DNNs

We present a differentiable formulation of diagonally sparse matrices (Sec. 3.1) to learn the pattern shown in Fig. 2. We then describe our $\mathrm{TopK}$ -based training approach (Sec. 3.2) that dynamically optimizes diagonal positions during training. Finally, we detail the conversion of our diagonal matrices to the GPU-efficient BCSR format (Sec. 3.3). Fig. 3 breaks down the various stages of our training method into forward and backward passes, illustrating what makes our DST approach efficient.

### 3.1. Diagonal Sparsity Formulation

To formulate our weight matrix with diagonal sparsity, we first define how the positions of the diagonals are determined and how the trainable parameters along these diagonals are specified.

**Permutation Matrix.** A permutation matrix $P \in \mathbb{R}^{M \times N}$ is a binary matrix containing precisely one entry of $1$ per row and column, with all other entries equal to $0$. In our setting, we place these 1s along a diagonal specified by an offset $\mathrm{off}$ as described in Eqn. 1.

**Value Vector.** Let $V \in \mathbb{R}^{\max(M,N)}$ be a vector whose elements populate the diagonal entries of $W$ at positions indicated by $P$. The operator $\mathrm{diag}(V)$ forms a diagonal matrix $K \times K$, where $K = \max(M, N)$, and places the elements of $V$ along its main diagonal.

**Diagonal Definition.** Let $W \in \mathbb{R}^{M \times N}$ and define $N = \min(M, N)$. We specify a diagonal in $W$ with offset $\mathrm{off}$ as the set of entries $(i, j)$ such that

$$j = (i + \mathrm{off}) \bmod N. \tag{1}$$

where a negative $\mathrm{off}$ indicates a diagonal below the main diagonal.

We aim to learn both the positions and the values of diagonals in $W$. To achieve this, we express $W \in \mathbb{R}^{M \times N}$ as the product of a permutation matrix $P$ and a diagonal matrix $\mathrm{diag}(V)$. Concretely, for a single diagonal weight matrix,

$$W_1 = P \times \mathrm{diag}(V) \tag{2}$$

This factorization enables gradient-based methods to optimize the corresponding values of $W$ (through $V$), making it well-suited for end-to-end learning.

We generalize the single-diagonal form in (2) to represent any matrix $W_K \in \mathbb{R}^{M \times N}$ with $K$ being the required number of diagonals (calculated from the desired sparsity level $S$[1]), each of length $\min(M, N)$. Specifically, we write

$$W_K = \sum_{j=1}^{K} P_j \, \mathrm{diag}(V_j), \tag{3}$$

where each $P_j$ is a permutation matrix defining the position of the $j$-th diagonal, and each $V_j$ is a vector of diagonal values.

### 3.2. TopK Based Diagonal Selection

With our learnable diagonal representation (Eqn. 3), we can parameterize each layer, and our objective reduces to finding the optimal diagonal placements (offsets) and associated values that yield the lowest overall loss.

---

[1] $K = \frac{(1-S) \cdot M \cdot N}{\min(M, N)}$

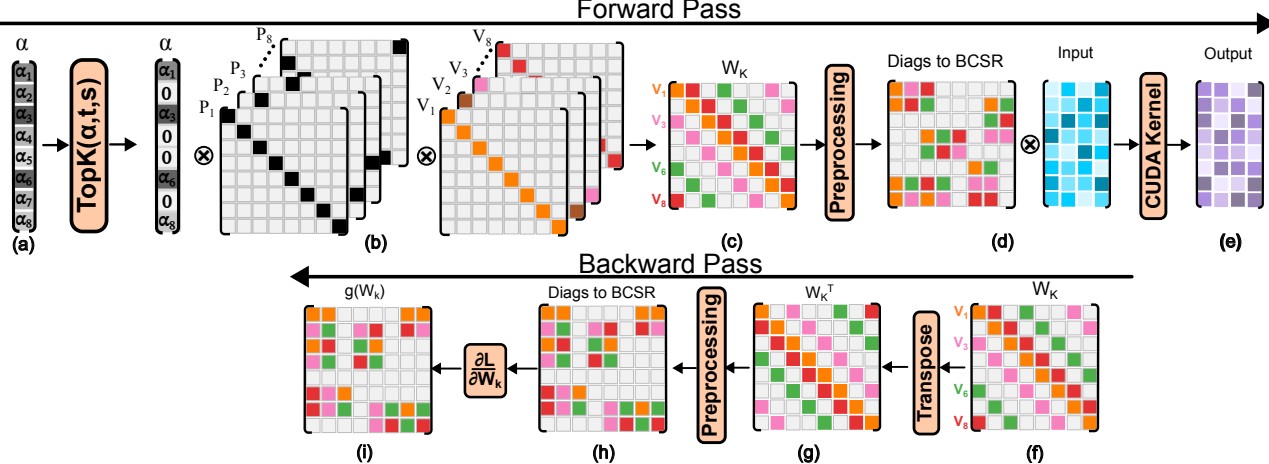

Figure 3: Training with DynaDiag and diagonal sparsity. TopK induces sparsity in $\alpha$ (a), which leads to selecting a subset of diagonals for the forward pass (b) and (c). Matrices with diagonals are converted to BCSR format to accelerate sparse matrix-matrix multiplication (d) and (e). Backward pass is accelerated by converting the diagonal sparse weight matrix to BCSR format (g). Sparse gradients are calculated using our custom CUDA kernels (h) and (i).

To determine which diagonals contribute the most to the final matrix $W_K$, we introduce a learnable vector of importance weights $\boldsymbol{\alpha} \in \mathbb{R}^{\max(M,N)}$ (Fig. 3a). We use a TopK function to select the $K$ most significant diagonals out of all possible diagonals based on the values in $\alpha$ (Fig. 3b). Concretely, we replace the summation in (3) with

$$W_K = \sum_{j=1}^{K} \tilde{\alpha}_j \, P_j \, \mathrm{diag}(V_j),$$

$$\tilde{\alpha} = \mathrm{TopK}(\boldsymbol{\alpha}), \qquad (4)$$

where $P_j$ and $\mathrm{diag}(V_j)$ specify the $j$-th diagonal in $W_K$, and $\tilde{\alpha}_j$ is the importance weight for that diagonal after applying TopK.

A differentiable TopK function takes a vector $\alpha$ of values as inputs and outputs a vector with higher values assigned to the top $K$ values. We can use a differentiable TopK function in an end-to-end training method and learn which values in $\alpha$ should be in the top $K$ for a particular task. We use the following softmax-based TopK in our end-to-end pipeline (we also experimented with other TopK methods, such as the one proposed by Sander et al. (2023) but found it to be too slow). For a given $\boldsymbol{\alpha} \in \mathbb{R}^{\max(M,N)}$ vector,

$$\tilde{\alpha}_i = \min\!\left( k \cdot \frac{\exp\!\left(\frac{\alpha_i}{T}\right)}{\sum_{j=1}^{n} \exp\!\left(\frac{\alpha_i}{T}\right)}, 1 \right) \qquad (5)$$

where $T$ is the temperature. We adopt a temperature-controlled TopK function to enable a balance between exploration (considering less dominant diagonals) and exploitation (focusing on the most important diagonals). Sub-optimal solutions may arise without temperature-controlled

TopK or heuristics, especially when the initial selection is biased or the significance of the diagonal evolves during training. We employ a cosine-annealing schedule during training to adjust $T$ from a high starting value (yielding a smoother TopK).To encourage sparsity in $\boldsymbol{\alpha}$, we employ an $\ell_1$ regularization term. We derive per-layer sparsity budget $\rho_j$ from the global sparsity budget $\rho_{\text{global}}$, and the number of diagonals $K_j$ for each layer is determined based on $\rho_j$.

Having established the forward pass with diagonal sparse matrices, we next describe how they are efficiently accelerated on GPUs by converting them into hardware-friendly formats.

### 3.3. GPU Acceleration

To efficiently execute diagonally sparse matrices on GPUs, they must be transformed into formats compatible with specialized hardware, such as dense matrices, 2:4 sparse matrices, or block sparse matrices. We determined that converting to block sparse matrices is the most effective approach for our sparse pattern (Fig. 3d): converting to dense matrices would introduce unnecessary overhead from additional vector addition operations, while 2:4 sparsity would impose overly restrictive constraints on the positioning of the diagonals.

When converting diagonal patterns to BCSR format, we optimize for two key objectives: minimizing the number of blocks and maximizing block density. Dense blocks ensure efficient hardware utilization by reducing redundant computations on zero values, while fewer blocks minimize memory access overhead and computational requirements on the GPU.

Although finding the optimal block-minimizing permutation from diagonals to BCSR is NP-hard, various heuristics have been developed to cluster non-zero values and reduce the block count. Our approach builds upon SmaT library, proposed by Okanovic et al. (2024). The SmaT library uses Jaccard-based similarity metric (Labini et al., 2022) for determining blocking, which works the best when matrices have a banded structure.

Our approach yields training acceleration by leveraging the same diagonal-to-BCSR conversion for $W_K$ and $W_K^{\mathsf{T}}$. This optimization enables efficient sparse computation during forward propagation and backpropagation, where we compute sparse gradients and accelerate the sparse operations required for gradient calculation.

Our hardware implementation also leverages key optimizations for sparse matrix multiplication on GPUs, inspired by the techniques described in (Okanovic et al., 2024). We elaborate on these techniques in more detail in Apdx. D.

## 4. Experiments

We first describe our experimental setup in Sec. 4.1. We then present our main results, comparing the performance of DynaDiag with other baselines (Sec. 4.2.1) and (Sec. 4.2.2) and comparing DST methods' training and inference time on GPUs (Sec. 4.2.3). We wrap up this section by proposing a fine-tuning method to improve DynaDiag's algorithmic performance beyond unstructured sparsity (Sec. 4.3.1).

### 4.1. Experimental Setup

The aim for our experiments is to show the efficacy of our structured DST approach on different modalities (vision and language), model types (mlp and attention), and sparsity regimes.

**Evaluation.** We assess the algorithmic performance of all training methods using Top-1 accuracy for vision tasks and perplexity for language tasks. Additionally, we measure the training and inference speedups achieved by each method at varying sparsity levels through end-to-end execution. We conduct paired asymptotic McNemar tests ($\alpha = 0.05$) comparing the top-performing method at each sparsity level against all others, bolding results that show no statistical difference from the best.

**Baselines.** We compare our approach against the following sparse training methods:

- **RigL** (Evci et al., 2020), MEST (Yuan et al., 2021), SET (Mocanu et al., 2018), CHT (Zhang et al., 2024), and CHTs (Zhang et al., 2025) uses DST to produce unstructured sparsity, which does not yield significant speedups in training or inference.

- **SRigL** (Lasby et al., 2023), **DSB** (Jiang et al., 2022), and **DiagHeur** train networks with structured sparsity via DST. SRigL exploits N:M sparsity to accelerate inference (but not training), DSB accelerates both, and DiagHeur serves as a diagonal-sparsity baseline without our differentiable topk selection (it uses a magnitude-based grow-and-decay scheme as detailed in Apdx. H).

- **PixelatedBFly** (Dao et al., 2021) factorizes dense matrices into a fixed butterfly structure. With further optimizations, it leverages block sparsity to speed up both training and inference.

### 4.2. Main Results

#### 4.2.1. VISION EXPERIMENTS

**Setup.** We evaluate two architectures for vision tasks on CIFAR10 and CIFAR100 (Krizhevsky & Hinton, 2009) and ImageNet-1K (Deng et al., 2009). Due to space limitations, we focus our discussion on ImageNet-1K results. For CIFAR10 and CIFAR100 results, please see Apdx. F.1.

- **MLP:** We use MLP-Mixer (Tolstikhin et al., 2021) to focus on the impact of sparsity on large matrix multiplication components without additional complexities.

- **ViT:** To demonstrate scalability, we train various sizes of Vision Transformers (ViT) (Dosovitskiy, 2020).

We test DynaDiag with uniform sparsity (Dao et al., 2021) at 60%, 70%, 80%, 90%, and 95%, following the training regime in Apdx. C.[2]

**Results on ImageNet-1K.** Tbl. 1 shows the performance of MLP-Mixer and three sparse variants of ViTs—small (S), large (L), and huge (H)—on ImageNet-1K. Except for ViT-L/16 and ViT-H/14, all other experiments are performed *three* times with average accuracies reported in the table.

The results highlight the effectiveness of DynaDiag, particularly at higher sparsities, where it consistently outperforms other structured sparse training (DST) methods. DynaDiag demonstrates significant improvements over competing DST methods at higher sparsities (90% and 95%). For instance: On ViT-L/16, DynaDiag achieves 77.74% accuracy at 90% sparsity, outperforming the next best method (SRigL) by 2.28%.To the best of our knowledge, we are the first to show a DST method's performance training large and huge variants of ViTs from scratch. DynaDiag consistently achieves results on par with RigL, SET, MEST, CHT and CHTs across most sparsity levels. We explain the reason behind

---

[2]All modules in ViT-S/16 are set to the desired sparsity level, except the multi-headed attention input projections.

Table 1: Top-1 accuracy of DynaDiag alongside baseline methods at varying sparsities. We bold results that are not significantly different (based on paired asymptotic McNemar tests ($\alpha = 0.05$)) from the best-performing method (marked with a *) in each column. Among all structured sparse training methods, DynaDiag achieves the highest accuracy on the ImageNet-1K dataset.

| Model | Method | Struc. | 60% | 70% | 80% | 90% | 95% |
|---|---|---|---|---|---|---|---|
| | | | *dense accuracy = 78.5* | | | | |
| | RigL | no | **79.75** | **79.28** | **78.71** | 77.24 | **71.50** |
| | SET | no | **78.15** | 78.01 | 77.78 | 77.01 | **71.48** |
| | CHT | no | **79.78** | **79.37** | 79.06* | 77.66* | 71.68* |
| **ViT-B/16** | CHTs | no | 79.88* | 79.38* | 79.05 | 77.54 | 71.61 |
| | MEST | no | **78.04** | 77.76 | 77.39 | 76.45 | 69.67 |
| | SRigl | yes | 77.79 | **77.84** | 77.35 | 75.90 | 68.70 |
| | PixelatedBFly | yes | **78.04** | 77.90 | 77.31 | 73.89 | 62.52 |
| | DSB | yes | 77.98 | 77.85 | 76.26 | 72.89 | 64.17 |
| | DiagHeur. | yes | 77.37 | 76.95 | 75.75 | 71.46 | 68.06 |
| | **DynaDiag** | yes | **78.29** | **77.94** | 77.62 | **76.91** | 69.54 |
| | | | *dense accuracy = 82.2* | | | | |
| | RigL | no | 81.85* | 81.57* | 81.7* | 78.26* | 72.11* |
| | SRigL | yes | 79.87 | 78.94 | 77.54 | 75.46 | 66.68 |
| **ViT-L/16** | PixelatedBFly | yes | 79.13 | 79.06 | 79.33 | 75.12 | 66.59 |
| | DSB | yes | 79.44 | 77.46 | 75.34 | 73.55 | 66.77 |
| | DiagHeur. | yes | 80.02 | 80.07 | 78.23 | 74.24 | 68.79 |
| | **DynaDiag** | yes | **81.52** | **81.56** | **81.37** | **77.74** | 69.59 |
| | | | *dense accuracy = 83.2* | | | | |
| | RigL | no | 83.84* | 83.47* | 82.85* | 80.52* | 74.54* |
| | SRigL | yes | 80.62 | 79.09 | 75.42 | 76.58 | 69.24 |
| **ViT-H/14** | PixelatedBFly | yes | **82.24** | 82.10 | 81.37 | 79.91 | 70.21 |
| | DSB | yes | 79.53 | 76.44 | 71.32 | 68.09 | 62.86 |
| | DiagHeur. | yes | 80.64 | 78.97 | 73.07 | 75.04 | 65.45 |
| | **DynaDiag** | yes | **82.76** | **82.60** | **81.89** | **80.44** | 73.74 |
| | | | *dense accuracy = 72.4* | | | | |
| | RigL | no | 73.21* | 73.23* | 73.47* | 73.01* | 70.41* |
| | SRigL | yes | **71.89** | 72.05 | 71.71 | 70.21 | 66.87 |
| **Mixer-S/16** | PixelatedBFly | yes | 71.95 | 71.91 | 71.45 | 69.17 | 67.85 |
| | DSB | yes | 69.94 | 70.21 | 68.9 | 65.16 | 60.88 |
| | DiagHeur. | yes | 69.91 | 69.77 | 67.87 | 64.17 | 59.16 |
| | **DynaDiag** | yes | **72.92** | **72.95** | **73.05** | **72.31** | **68.89** |

this disparity in accuracy between DynaDiag and unstructured methods in Sec. 4.3.1 to surpass RigL's performance.)

DynaDiag achieves superior accuracy than other DST approaches on all the ViT variants, demonstrating the scalability of our method to large models while maintaining competitive performance. The p-values under the asymptotic McNemar's test are reported in Tbl. 10 in Apdx. E. This trend holds for CIFAR10 and CIFAR100 experiments as well.

### 4.2.2. LANGUAGE EXPERIMENTS

**Setup.** For language tasks, we evaluate on the WikiText-103 dataset using two variants (Small and Medium) of GPT-2 (Radford et al., 2019) at varying sparsities $s \in \{40\%, 50\%, 60\%, 80\%, 90\%\}$. This setup aims to reduce

the substantial memory and computational overheads associated with increasing parameter counts in large language models.

**Results on WikiText-103.** We train GPT2-Small and GPT2-Medium (Radford et al., 2019) from scratch[3] on WikiText-103 data set and show the resulting performance in Tbl. 2. All experiments are performed *two* times with average accuracies reported in the table. Both GPT2 models trained with DynaDiag outperform other DST methods on the perplexity metric (lower the better).

Notably, DynaDiag demonstrates increasingly significant improvements over competing DST approaches as sparsity levels rise. For example, at 90% sparsity on GPT2-Small,

---

[3]We make both the attention and MLP layers sparse.

DynaDiag achieves a perplexity of 6.22 points lower than that of the next best DST method while remaining within 2.57 points of the performance ceiling established by RigL. The p-values under the asymptotic McNemar's test are reported in Tbl. 11 in Apdx. E.

Table 2: Perplexity of DynaDiag and baselines. We bold results that are not significantly different from RigL based on paired asymptotic McNemar tests ($\alpha = 0.05$). Among all the baselines, DynaDiag achieves the lower PPL (lower, the better) on the WikiText-103 dataset.

| Model | Method | 40% | 50% | 60% | 80% | 90% |
|---|---|---|---|---|---|---|
| | | *dense accuracy = 22.21* | | | | |
| GPT2-S | RigL | 22.34 | 22.80 | 23.79 | 29.87 | 53.76 |
| | SRigL | 22.74 | 23.19 | 25.09 | 31.08 | 62.55 |
| | PixelatedBFly | **22.50** | 23.25 | 25.98 | 34.89 | 66.44 |
| | **DynaDiag** | 22.60 | **22.74** | **24.67** | **30.46** | 56.33 |
| | | *dense accuracy = 20.18* | | | | |
| GPT2-M | RigL | 20.45 | 21.60 | 23.49 | 28.87 | 51.76 |
| | SRigL | 21.14 | 22.59 | 26.09 | 32.16 | 55.66 |
| | PixelatedBFly | **20.86** | 22.49 | 25.45 | 34.24 | 56.09 |
| | **DynaDiag** | **20.69** | **22.14** | **24.98** | **29.65** | 54.87 |

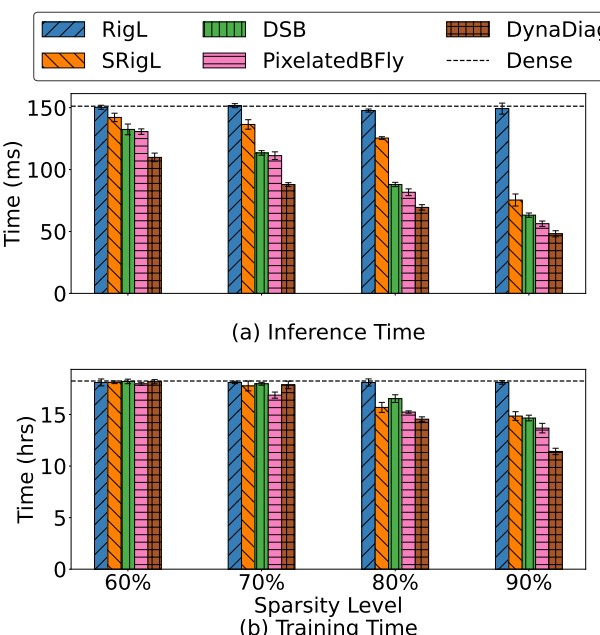

(a) Inference Time

(b) Training Time

Figure 4: Inference and training time of a ViT-Base at different sparsity levels trained with DynaDiag and baselines. When controlling for an equivalent number of non-zero parameters, our specialized diagonal-sparse kernel achieves faster execution on GPUs than standard sparse kernels.

### 4.2.3. TRAINING AND INFERENCE ACCELERATION

We measure GPU training and inference times for the methods described in Sec. 4.1. All reported timings reflect wall-clock measurements.

**Setup.** We select the most suitable libraries to accelerate execution based on the structured sparsity type: (1) *RigL:* We use CUSPARSE, as RigL lacks exploitable structure. (2) *SRigL:* The authors propose a custom method for the acceleration of inference in SRigL (Lasby et al., 2023), which we employ to estimate inference and training times. However, the provided CUDA kernels lack the necessary supporting infrastructure (e.g., integration with PyTorch[4]) to enable end-to-end execution and real-time data collection. (3) *DSB and PixelatedBFly:* Both methods yield block-sparse weight matrices. We use the Triton-based library from PixelatedBFly (Dao et al., 2021) to accelerate inference for both. However, while this library is used for PixelatedBFly training, it cannot be applied to DSB due to the significant effort required to integrate it into DSB's training pipeline. (4) *Diag:* We employ custom-developed kernels to accelerate both training and inference for diagonal sparse networks.

**Training & Inference Time.** In Fig. 4, we present real-world timing comparisons of DynaDiag and other dynamic sparse training (DST) approaches. We implement custom layers in PyTorch for each method and leverage their dedicated CUDA kernels to accelerate execution. Our results demonstrate that DynaDiag achieves an inference speedup of up to $3.1\times$ and training speedup of $1.59\times$ compared to the dense equivalent at 90% sparsity. However, we observe diminishing gains at lower sparsity levels, with inference speedups of $1.37\times$ at 60% sparsity and training times approaching parity ($0.98\times$ at 60% sparsity). Our PyTorch implementation does not exploit CUDA kernel optimizations, and addressing this could lead to further speed increases.

### 4.3. Additional Results

#### 4.3.1. HOW TO IMPROVE DYNADIAG PERFORMANCE?

Tbl. 1 reveals a performance gap between RigL and DynaDiag at sparsities $\geq 80\%$, which we attribute to the increased expressivity of unstructured sparsity in RigL (Liu & Wang, 2023). To validate this hypothesis, we conduct an experiment using LoRA-FA (Zhang et al., 2023a) to fine-tune the weight matrices of a ViT-B/16 model at 80% sparsity trained with DynaDiag. We choose LoRA-FA specifically for its memory and computational efficiency during fine-tuning.

As shown in Fig. 5a), increasing the rank of LoRA-FA's A and B matrices leads to improved model accuracy. No-

---

[4]While custom CUDA kernels can theoretically accelerate training, their effectiveness depends on the surrounding infrastructure, such as memory management and integration with deep learning frameworks like PyTorch. In DSB and SRigL, the absence of this supporting structure limits their practical utility for end-to-end training acceleration.

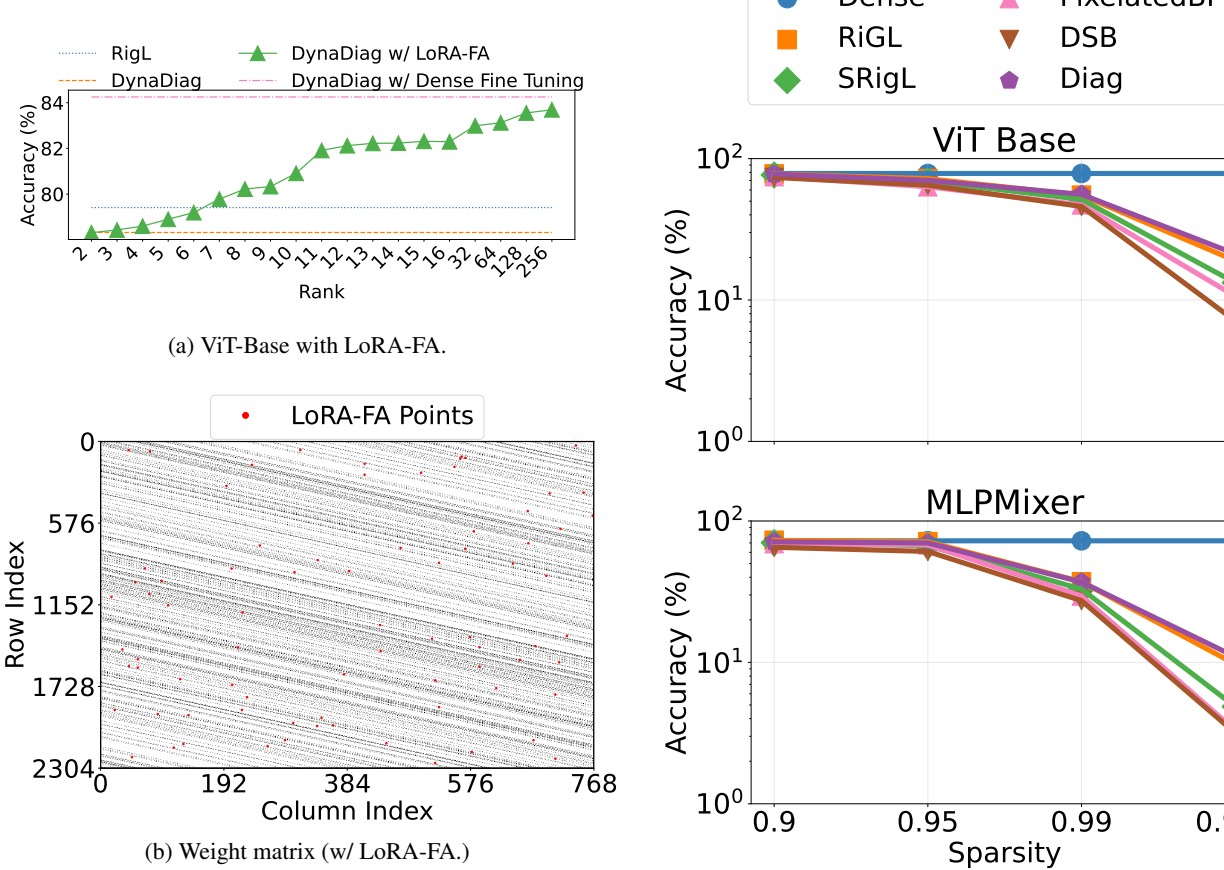

(a) ViT-Base with LoRA-FA.

(b) Weight matrix (w/ LoRA-FA.)

Figure 5: Performance of ViT-Base at 80% sparsity fine-tuned with LoRA-FA matrices of different ranks (a). We see an increase in the accuracy over a diagonally sparse model, with rank six matrices enough to match RigL performance. (b) Shows a distribution of the new fine-tuned parameters for a linear layer in the attention block of ViT-Base.

Figure 6: Performance of DynaDiag compared to other baselines at extreme sparsities for a ViT-Base and MLPMixer models on ImageNet-1K. DynaDiag outperforms other baselines, including RigL, at extreme sparsities.

tably, DynaDiag surpasses RigL's performance at rank 6, requiring only a 1.67% increase in total parameters. This minimal parameter overhead demonstrates that DynaDiag can exceed the accuracy ceiling of unstructured sparsity while maintaining most of its structured sparsity benefits for inference acceleration.

We examine the spatial distribution of fine-tuned parameters in Fig. 5, analyzing a linear layer within an attention head of ViT-B/16 trained on ImageNet-1K. The visualization reveals that the newly introduced parameters are distributed across the weight matrix in an unstructured pattern, supporting our hypothesis that unstructured sparsity enables better optimization of the loss landscape.

### 4.3.2. PERFORMANCE AT EXTREME SPARSITY

Dynamic sparse training typically uses predefined layerwise sparsity ratios (e.g., uniform) to avoid layer collapse at high sparsity levels, where entire layers risk being pruned (Tanaka et al., 2020). However, individual layers exhibit distinct learning dynamics (Chen et al., 2023), and extreme sparsity can disrupt gradient flow. We evaluate DynaDiag under sparsity levels of up to 99.99% on ImageNet-1K using ViT-Base and MLP-Mixer. As shown in Fig. 6, DynaDiag demonstrates robust performance even under these extreme conditions, indicating that it effectively discovers accurate masks from the network topology—even in highly sparse settings. DynaDiag outperforms RigL at extreme sparsities, where RigL's performance is known to degrade significantly, as documented in prior work (Ji et al., 2024).

# 5. Discussion

We have identified three current limitations of DynaDiag, which will be a significant part of the future work. First, while effective for ViTs and LLMs, DynaDiag faces scalability challenges with CNNs due to the overhead of searching for distinct diagonal patterns across each channel. Secondly, we aim to extend our method to networks with extremely sparse weight matrices—e.g., at sparsity levels approaching 99.9999% or more—while still retaining more than a single active diagonal. We believe that with large matrices, we will see the effectiveness of having small-world connectivity, resulting in highly sparse networks. However, we could not find a good example model to test our hypothesis. Finally, our method's performance could be further improved through optimized GPU implementations of diagonal sparsity using frameworks like Triton.

# 6. Conclusion

This work presents a novel dynamic sparse training algorithm, DynaDiag, that uses small world networks inspired diagonal structured sparsity to train neural networks. We demonstrate that models trained using our TopK -based algorithm surpass those from other structured sparse training approaches in terms of both algorithmic effectiveness and hardware acceleration. Ultimately, we show that DynaDiag achieves competitive performance — matching other unstructured sparsity training methods at lower sparsity levels while beginning to match and, in some cases, surpassing them at extreme sparsity levels.

# Acknowledgments

This work was partly supported by NSF award #2326491 to CK and #2419721 to WR and YZ. The views and conclusions contained herein are those of the authors and should not be interpreted as representing any sponsor's official policies or endorsements.

# Impact Statement

Our work introduces an efficient sparse training method that enables training larger models with fixed computational resources, supporting the democratization of deep learning as model sizes continue to grow. The method's ability to maintain sparsity throughout training reduces computational and energy costs during both training and deployment, particularly benefiting on-device and real-time learning scenarios. While we anticipate primarily positive impacts through improved resource efficiency and accessibility, we acknowledge that any advancement in machine learning capabilities warrants careful consideration. We encourage future research to investigate potential differential effects of our sparsification approach on model fairness and bias, particularly regarding class imbalances and data distribution shifts compared to dense models.

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

# Appendix

## A. Proving the Transposability of Diagonal Sparse Matrices

THEOREM

Let $M \in \{0,1\}^{m \times n}$ be a binary mask matrix constructed with a pseudo-diagonal pattern starting at position $s$. Then, the transpose $M^\top$ also exhibits a pseudo-diagonal pattern, with the diagonal's starting position adjusted based on the relative dimensions of $m$ and $n$.

PROOF

We consider two cases based on the relationship between $m$ and $n$:

CASE 1: $m \geq n$ (MORE ROWS THAN COLUMNS)

1. **Construction of $M$:**

   - The diagonal starts at row $s$ and spans $n$ entries.
   - Entries in $M$ are at positions:

   $$(r, c) = ((s + c) \mod m, \, c) \quad \text{for } c \in \{0, 1, \ldots, n - 1\}.$$

2. **Transposition $M^\top$:**

   - The transposed matrix $M^\top$ has dimensions $n \times m$.
   - Entries in $M^\top$ are at positions:

   $$(c, (s + c) \mod m) \quad \text{for } c \in \{0, 1, \ldots, n - 1\}.$$

3. **Pseudo-Diagonal in $M^\top$:**

   - Since $n \leq m$, the transposed matrix $M^\top$ now has fewer rows than columns.
   - The pseudo-diagonal in $M^\top$ starts at **column** $s$ (original row offset) and spans $n$ entries:

   $$(r, c) = (r, \, (s + r) \mod m) \quad \text{for } r \in \{0, 1, \ldots, n - 1\}.$$

   - This matches the transposed entries, confirming $M^\top$ retains a pseudo-diagonal starting at column $s$.

CASE 2: $m < n$ (MORE COLUMNS THAN ROWS)

1. **Construction of $M$:**

   - The diagonal starts at column $s$ and spans $m$ entries.
   - Entries in $M$ are at positions:

   $$(r, c) = (r, \ (s + r) \mod n) \quad \text{for } r \in \{0, 1, \ldots, m - 1\}.$$

2. **Transposition $M^\top$:**

   - The transposed matrix $M^\top$ has dimensions $n \times m$.
   - Entries in $M^\top$ are at positions:

   $$((s + r) \mod n, \ r) \quad \text{for } r \in \{0, 1, \ldots, m - 1\}.$$

3. **Pseudo-Diagonal in $M^\top$:**

   - Since $n > m$, $M^\top$ has more rows than columns.
   - The pseudo-diagonal in $M^\top$ starts at **row** $s$ (original column offset) and spans $m$ entries:

   $$(r, c) = ((s + c) \mod n, \ c) \quad \text{for } c \in \{0, 1, \ldots, m - 1\}.$$

   - This aligns with the transposed entries, confirming $M^\top$ retains a pseudo-diagonal starting at row $s$.

In both cases, transposing a pseudo-diagonal mask results in a matrix with an equivalent pseudo-diagonal structure. The starting position $s$ transitions between row and column offsets depending on the original matrix's dimensions, preserving the diagonal nature under transposition. Thus, matrices constructed with the described pseudo-diagonal pattern exhibit transposition invariance in their sparsity structure.

## B. Theoretical Justification of Diagonal Sparsity

Consider a single linear layer of a neural network represented by the weight matrix $W \in \mathbb{R}^{n \times m}$. Sparsity patterns enforce structural constraints on $W$ through binary masks $M \in \{0, 1\}^{n \times m}$, producing sparse weight matrices as $W \odot M$, where $\odot$ denotes element-wise multiplication.

**Definition (Diagonal Sparsity).** A *diagonal sparsity pattern* selects a set of diagonals defined by their offsets. Formally, the mask $M$ is constructed as:

$$M_{ij} = \begin{cases} 1, & \text{if } (j - i) \equiv \delta \ (\text{mod } \min(n, m)) \text{ for some diagonal offset } \delta, \\ 0, & \text{otherwise.} \end{cases}$$

The number of diagonals selected, denoted $k$, controls the density of $M$.

**Lemma 1 (Full Input-Output Coverage).** A diagonal mask $M$ constructed as above ensures that each row and each column of $M$ contains at least one nonzero entry, given $k \geq 1$.

**Proof.** Consider two cases:

- **Case $n \geq m$:** Each diagonal covers all $m$ columns exactly once. By varying diagonal offsets, each row index $i$ is represented by some diagonal at least once, thus no row remains empty. Since each diagonal always covers every column exactly once, no column remains empty either.

- **Case $m > n$:** By symmetry, each diagonal covers all $n$ rows exactly once. By shifting offsets across multiple diagonals, all columns are covered at least once.

This ensures no input dimension or output neuron is ever completely disconnected by the sparsity mask, guaranteeing full coverage. □

**Expressivity via Universal Approximation.** The universal approximation theorem (Cybenko, 1989; Hornik, 1991) asserts that fully connected neural networks with nonlinear activations (such as ReLU or sigmoid) are capable of approximating arbitrary continuous functions. Structural sparsity patterns, however, may potentially violate universal approximation by severing input-output connections.

**Theorem 2 (Universal Approximation under diagonal Sparsity).** A feed-forward neural network consisting of layers employing diagonal sparsity masks with sufficiently large number of diagonals $k \geq k_{\min}$, nonlinear activations, and adequate width and depth, retains the universal approximation property.

**Proof** Universal approximation requires that any input neuron can influence any output neuron, possibly through multiple layers. Lemma 1 ensures full input-output coverage at every sparse linear transformation. Thus, each neuron maintains at least one active input and output edge, preserving connectivity across the network.

Because diagonal patterns ensure no dimension is eliminated globally, subsequent nonlinear activations can still combine these preserved dimensions arbitrarily. Therefore, the assumptions required for universal approximation [1][2] remain valid, implying the network's expressivity is not significantly impaired by this structured sparsity pattern. □

**Rank Preservation Argument.** Another perspective on expressivity arises through linear algebraic arguments. Any row or column in a weight matrix $W$ being entirely zero automatically restricts the maximum achievable rank. A rank-deficient weight matrix severely limits the set of linear transformations expressible by that layer.

With diagonal sparsity:

- Each row and column has at least one nonzero element, removing a trivial structural cause of rank deficiency.

- For random initialization of nonzero weights, such sparse matrices will have full achievable rank ($\min(n, m)$) almost surely (standard linear algebra argument; see e.g. random matrix theory.).

High rank at initialization and throughout training supports richer linear transformations, ultimately preserving higher network expressivity and better gradient propagation properties.

**Comparison to Other Patterns.** Unlike structured sparsity patterns such as block-sparse or $n$-of-$m$ sparsity, diagonal sparsity guarantees deterministic full input-output coverage. In block sparsity, some neurons risk being completely disconnected if blocks overlap in suboptimal ways, limiting expressivity and training stability. diagonal sparsity patterns avoid these degeneracies by design.

Furthermore, diagonal sparsity provides a structured yet near-random pattern, capturing many beneficial properties of unstructured random sparsity, demonstrated empirically by experiments in this paper—while offering improved computational efficiency and implementation simplicity.

These theoretical results substantiate the empirically observed superior or near-optimal performance of diagonal sparsity patterns, positioning them as a particularly promising structured sparsity approach for efficient neural network training and inference.

## C. Experiment Details

All experiments are conducted on the NVIDIA Tesla A100 GPUs with the following configuration:

- Model: NVIDIA A100 80GB

- Memory: 80GB HBM2e

- Memory Bandwidth: ∼2.0 TB/s (higher than the 40GB version)

- TDP : 400W (PCIe: 300W)

- Peak FP32 Performance: $\sim$19.5 TFLOPS (same as 40GB)

- Peak FP16 Performance: $\sim$312 TFLOPS (same as 40GB)

### C.1. Datasets

1. **CIFAR-10** (Krizhevsky & Hinton, 2009) consists of 60,000 colored images of resolution $32 \times 32$, divided into 10 classes (e.g., airplanes, cars, birds). The dataset is split into 50,000 training and 10,000 test images.

2. **CIFAR-100** (Krizhevsky & Hinton, 2009) also contains $32 \times 32$ resolution images but spans 100 classes. Each class includes 500 training and 100 test images, totaling 60,000 images.

3. **ImageNet-1K** (Deng et al., 2009) covers 1,000 object classes, with 1.28M training, 50,000 validation, and 100,000 test images. Images are typically resized and cropped to $224 \times 224$ for processing.

4. **WikiText-103** (Merity et al., 2016) comprises over 100 million tokens extracted from verified Wikipedia articles. It is significantly larger than other language datasets, such as Penn Treebank (PTB) (Marcus et al., 1993).

Table 3: Configuration of the CIFAR10 and CIFAR100 experiments with MLPMixer.

| Parameter | Value |
| --- | --- |
| Adam $\beta_1$ | 0.9 |
| Adam $\beta_2$ | 0.99 |
| AutoAugment | True |
| Batch Size | 128 |
| CutMix Probability | 0.5 |
| CutMix $\beta$ | 1.0 |
| Dropout | 0.0 |
| Epochs | 300 |
| Hidden_C | 512 |
| Hidden_S | 64 |
| Hidden | 128 |
| (Initial LR, Final LR) | $(1 \times 10^{-3}, 1 \times 10^{-6})$ |
| Label Smoothing | 0.1 |
| Layers | 8 |
| LR Scheduler | Cosine |
| Optimizer | Adam |
| Random Seed | 3407 |
| Weight Decay | $5 \times 10^{-5}$ |
| Warmup | 5 epochs |

Table 4: Configuration of the CIFAR10 and CIFAR100 experiments with ViT-Small.

| Parameter | Value |
| --- | --- |
| Epochs | 200 |
| Batch Size | 128 |
| Optimizer | Adam |
| Weight Decay | $5 \times 10^{-5}$ |
| LR Scheduler | Cosine |
| (Initial LR, Final LR) | $(1 \times 10^{-3}, 1 \times 10^{-5})$ |
| Warmup | 5 epochs |
| Dropout | 0.0 |
| AutoAugment | True |
| Label Smoothing | 0.1 |
| Heads | 12 |
| Layers | 7 |
| Hidden | 384 |
| MLP Hidden | 384 |

Table 5: Configuration of the ImageNet experiments with ViT-Base and MLPMixer.

| Model | Optimizer | Weight Decay | Learning Rate | Drop Path | Warmup/Epoch |
|---|---|---|---|---|---|
| ViT-Base | AdamW | 0.05 | 0.001 | 0.1 | 5/300 |
| DynaDiag-ViT-Base | AdamW | 0.05 | 0.001 | 0 | 5/300 |
| Mixer-Small | AdamW | 0.1 | 0.001 | 0.1 | 5/300 |
| DynaDiag-Mixer-Small | AdamW | 0.1 | 0.001 | 0 | 5/300 |

Table 6: Configuration of the ImageNet experiments with ViT-Large and Huge.

| Parameter | Value |
|---|---|
| Batch size | 256 |
| Optimizer | AdamW |
| Learning Rate (LR) | $3 \times 10^{-3}$ |
| LR decay | cosine |
| Weight decay | 0.02 |
| Warmup epochs | 5 |
| Label smoothing $\varepsilon$ | 0.1 |
| Dropout | ✗ |
| Stochastic Depth | ✓ |
| Repeated Aug | ✓ |
| Gradient Clipping | 1.0 |
| Horizontal flip | ✓ |
| Random Resized Crop (RRC) | ✓ |
| Rand Augment | ✗ |
| 3 Augment (ours) | ✓ |
| LayerScale | ✓ |
| Mixup $\alpha$ | 0.8 |
| Cutmix $\alpha$ | 1.0 |
| Erasing prob. | ✗ |
| ColorJitter | 0.3 |
| Test crop ratio | 1.0 |
| Loss | BCE |

## D. GPU Acceleration

We leverage the known diagonal structure to enforce the clustering of rows/columns from the same diagonal into contiguous blocks. Modify the reordering algorithm to:

- Prioritize grouping rows/columns from the same diagonal.

- Allow limited flexibility for rows/columns from adjacent diagonals (controlled by Diag_Proximity).

We modify Jaccard's similarity with a diagonal proximity term to prioritize clustering rows/columns that belong to diagonals with nearby starting positions. For two rows $i$ and $j$:

$$\text{Sim}(i, j) = \alpha \cdot \text{Jaccard}(i, j) + (1 - \alpha) \cdot \text{Proximity}(i, j) \tag{6}$$

where:

- Jaccard$(i, j)$: Standard Jaccard index (overlap of non-zeros between rows $i$ and $j$).

- Proximity$(i, j)$: Normalized inverse distance between the starting positions of the diagonals containing rows $i$ and $j$:

$$\text{Proximity}(i, j) = 1 - \frac{\text{dist}(d_i, d_j)}{\max(\text{dist})} \tag{7}$$

Table 7: Configuration of the Wikitext-103 experiments GPT-2Small experiments.

| Model | Optimizer | Weight Decay | Learning Rate | Dropout | Warmup/Epoch |
|-------|-----------|--------------|---------------|---------|--------------|
| GPT-2-Small | AdamW | 0.1 | 0.0001 | 0.1 | 5/100 |
| DynaDiag | AdamW | 0.1 | 0.0001 | 0.1 | 5/100 |

where $d_i, d_j$ are the diagonal start positions for rows $i, j$, and $\max(\text{dist})$ is the maximum possible distance between diagonals.

- $\alpha$: Tuning parameter ($0 \leq \alpha \leq 1$) to balance the two terms. For our case, we set $\alpha < 0.5$ to prioritize diagonal structure over raw overlap.

Since the diagonals are determined by their starting positions, we precompute the diagonal membership for each row/column. Using this preprocessing step, we can convert sparse diagonal matrices to dense blocks during the forward (Fig. 3d) and backward (Fig. 3h) pass which accelerates both inference and training of models with DynaDiag.

Specifically, we employ the following strategies to maximize computational efficiency and minimize memory overhead:

- **Tensor Cores (TC) API**: We utilize half-precision matrix multiply-accumulate (`mma.m16n8k16`) operations via the TC API, as illustrated in the PTX code provided in (Okanovic et al., 2024). This allows us to exploit the high throughput of Tensor Cores for dense submatrix computations within the sparse matrix framework.

- **Block Compressed Sparse Row (BCSR) Iteration**: To efficiently iterate over non-zero blocks in the sparse matrix, we use the BCSR format, which relies on `rowPtr` and `colIdx` arrays. This avoids unnecessary iterations over zero blocks, significantly reducing computational overhead.

- **Asynchronous Data Movement**: To hide memory latency, we employ `cuda::memcpy_async` for overlapping computation with data transfers. This allows efficient movement of data from global memory to shared memory without intermediate register staging, freeing up registers for computation.

- **Warp-Level Parallelism**: Each warp in our CUDA kernel is responsible for computing a submatrix of the output matrix $C$, with dimensions matching those of the Tensor Cores. Non-zero blocks are loaded into registers using `ldmatrix` (see Listings 2 and 3 in (Okanovic et al., 2024)), and the results are written back to global memory after computation.

For a comprehensive understanding of the implementation details, including pseudocode and additional optimizations, we direct readers to (Okanovic et al., 2024). The SMAT library can be found at https://github.com/spcl/smat.

**Correlation Between Number of Diagonals and Speedups.** Using our custom CUDA implementation, we perform matrix-matrix multiplication on matrices of size $768 \times 768$(matching the size of the blocks.I.attn.proj.linear.weight layer in ViT where I is the block count) to isolate the impact of number of diagonals on potential speedup. All the experiments were carried out on NVIDIA A100 40GB GPUs. Each configuration was run 100 times, and we averaged the total time of converting diagonals to BCSR plus the subsequent BCSR computation. As expected, below 50% sparsity, speed gains taper off, and below 20% sparsity, we see some slowdown—yet this remains more favorable than comparable block-sparse approaches(Yamaguchi & Busato, 2021).

**Performance Impact of Diagonals to BCSR Conversion.** With empirical experiments on ViT-B/16 at 90% sparsity with ImageNet-1K, we verify that there is no significant accuracy difference between direct diagonal computation and the BCSR-based approach. From Tbl. 8, we can see that there is no significant difference in the accuracies of the two methods, proving their equivalence. However, the training time is significantly improved using our custom BCSR kernel.

## E. McNemar's Test Results

McNemar's test is a non-parametric $\chi^2$ procedure for paired binary outcomes. Given two classifiers (or a pre–post condition) evaluated on the same $n$ instances, their predictions form a $2 \times 2$ contingency table. The statistic considers only the off-diagonal counts—instances mis-classified by exactly one of the two methods—and tests the null hypothesis that these two counts are equal, i.e., that the marginal proportions of successes are identical. A significant result therefore indicates

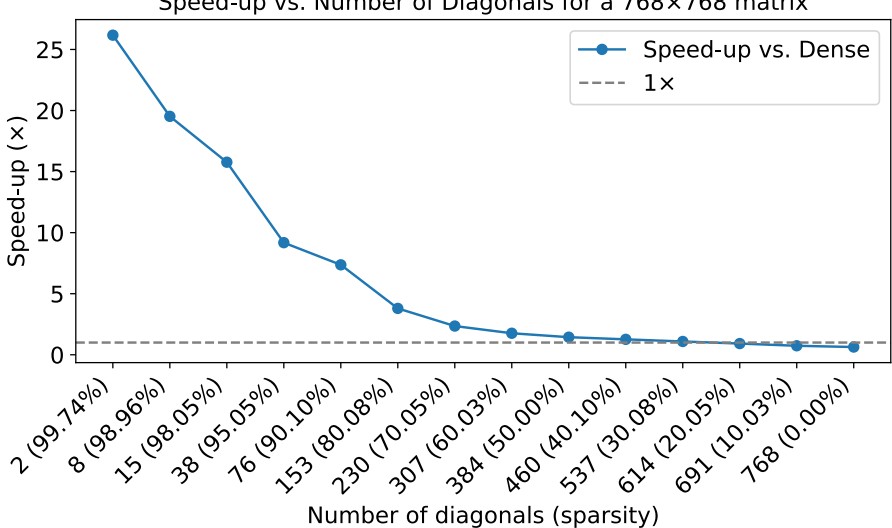

Figure 7: Speedup obtained using our custom CUDA implementation while doing matrix-matrix multiplication with matrices having the diagonal sparsity pattern.

Table 8: DynaDiag performance with and without BCSR conversion with a 90% sparse ViT-B/16 on ImageNet-1K.

| Method | Accuracy | Training Time (hrs) |
|---|---|---|
| Without BCSR Conversion | 76.92±0.0011 | 18.07 |
| With BCSR Conversion | 76.91±0.0007 | 11.42 |

Table 9: P-values from McNemar's test comparing each method with RigL at varying sparsities $s \in \{0.6, 0.7, 0.8, 0.9, 0.95\}$. Bold values indicate no significant difference (p-value $\geq 0.05$) from RigL.

| Models | Methods | Struc. | Cifar10 | | | | | Cifar100 | | | | |
|---|---|---|---|---|---|---|---|---|---|---|---|---|
| | | | 0.6 | 0.7 | 0.8 | 0.9 | 0.95 | 0.6 | 0.7 | 0.8 | 0.9 | 0.95 |
| | RigL | no | - | - | - | - | - | - | - | - | - | - |
| | SRigL | yes | **0.0523** | **0.0614** | 0.0312 | 0.0285 | 0.0221 | **0.0654** | **0.0531** | **0.0598** | 0.0356 | 0.0299 |
| Mixer-S/16 | PixelatedBFly | yes | **0.0751** | **0.0845** | 0.0213 | 0.0198 | 0.0174 | 0.0321 | 0.0212 | 0.0315 | 0.0287 | 0.0253 |
| | DSB | yes | 0.0231 | 0.0198 | 0.0123 | 0.0098 | 0.0075 | 0.0225 | 0.0187 | 0.0112 | 0.0085 | 0.0063 |
| | DiagHeur. | yes | 0.0145 | 0.0123 | 0.0091 | 0.0067 | 0.0052 | 0.0132 | 0.0110 | 0.0084 | 0.0058 | 0.0047 |
| | DynaDiag | yes | 0.0587 | **0.0674** | **0.0512** | **0.0543** | **0.0601** | **0.0756** | **0.0823** | **0.0654** | **0.0682** | **0.0745** |
| | RigL | no | - | - | - | - | - | - | - | - | - | - |
| | SRigL | yes | **0.0578** | **0.0695** | 0.0345 | 0.0302 | 0.0253 | **0.0712** | 0.0421 | **0.0623** | 0.0381 | 0.0324 |
| ViT-S/16 | PixelatedBFly | yes | 0.0245 | 0.0193 | 0.0167 | 0.0132 | 0.0111 | 0.0223 | **0.0676** | **0.0650** | 0.0125 | 0.0103 |
| | DSB | yes | 0.0184 | 0.0165 | 0.0122 | 0.0095 | 0.0078 | 0.0193 | 0.0161 | 0.0124 | 0.0090 | 0.0072 |
| | DiagHeur. | yes | 0.0123 | 0.0105 | 0.0087 | 0.0065 | 0.0051 | 0.0131 | 0.0114 | 0.0093 | 0.0069 | 0.0050 |
| | DynaDiag | yes | **0.0612** | **0.0725** | **0.0534** | **0.0568** | **0.0632** | **0.0773** | **0.0834** | **0.0675** | **0.0708** | **0.0761** |

that the two methods differ in predictive accuracy while properly accounting for the dependence induced by evaluating on the same data points.

We show the p-values for ViT-S/16 and Mixer-S/16 running Cifar10 and Cifar100 in Tbl. 9. And the p-values for models on ImageNet-1K and GPT models on WikiText-103 are shown in Tbl. 10 and Tbl. 11 respectively.

# F. Additional Results

## F.1. CIFAR10 and CIFAR100 Results

All the experiments are performed *three* times with the average accuracies shown in the table. We omit the standard deviation since the training is relatively stable (i.e., usually less than 0.06% standard deviation). We present results for Mixer-S/16

Table 10: P-values from paired asymptotic McNemar tests comparing each method with RigL at varying sparsities $s \in \{60\%, 70\%, 80\%, 90\%, 95\%\}$. Bold values indicate no significant difference (p-value $\geq 0.05$) from RigL.

| Model | Method | 60% | 70% | 80% | 90% | 95% |
|---|---|---|---|---|---|---|
| **ViT-B/16** | RigL | - | - | - | - | - |
| | SET | **0.0542** | 0.0401 | 0.0364 | 0.0419 | **0.0564** |
| | CHT | **0.0764** | **0.0672** | **0.0693** | **0.0628** | **0.0572** |
| | CHTs | **0.0654** | **0.0619** | **0.0629** | **0.0724** | **0.0529** |
| | MEST | **0.0598** | 0.0462 | 0.0298 | 0.0387 | 0.0432 |
| | SRigL | 0.0473 | **0.0521** | 0.0345 | 0.0289 | 0.0224 |
| | PixelatedBFly | **0.0542** | 0.0396 | 0.0317 | 0.0253 | 0.0198 |
| | DSB | 0.0124 | 0.0187 | 0.0105 | 0.0083 | 0.0061 |
| | DiagHeur. | 0.0098 | 0.0123 | 0.0075 | 0.0059 | 0.0043 |
| | **DynaDiag** | **0.0654** | **0.0789** | 0.0411 | **0.0598** | 0.0352 |
| **ViT-L/16** | RigL | - | - | - | - | - |
| | SRigL | 0.0415 | 0.0253 | 0.0367 | 0.0302 | 0.0235 |
| | PixelatedBFly | 0.0398 | 0.0312 | 0.0329 | 0.0264 | 0.0201 |
| | DSB | 0.0142 | 0.0201 | 0.0113 | 0.0091 | 0.0067 |
| | DiagHeur. | 0.0105 | 0.0137 | 0.0082 | 0.0065 | 0.0049 |
| | **DynaDiag** | **0.0698** | **0.0836** | **0.0564** | **0.0517** | 0.0379 |
| **ViT-H/14** | RigL | - | - | - | - | - |
| | SRigL | 0.0431 | 0.0395 | 0.0382 | 0.0320 | 0.0258 |
| | PixelatedBFly | **0.0627** | 0.0343 | 0.0341 | 0.0276 | 0.0214 |
| | DSB | 0.0165 | 0.0223 | 0.0131 | 0.0105 | 0.0079 |
| | DiagHeur. | 0.0128 | 0.0159 | 0.0095 | 0.0078 | 0.0056 |
| | **DynaDiag** | **0.0725** | **0.0874** | **0.0583** | **0.0529** | 0.0394 |
| **Mixer-S/16** | RigL | - | - | - | - | - |
| | SRigL | **0.0672** | 0.0294 | 0.0356 | 0.0294 | 0.0230 |
| | PixelatedBFly | 0.0311 | 0.0491 | 0.0308 | 0.0247 | 0.0185 |
| | DSB | 0.0139 | 0.0195 | 0.0102 | 0.0079 | 0.0054 |
| | DiagHeur. | 0.0112 | 0.0148 | 0.0087 | 0.0063 | 0.0045 |
| | **DynaDiag** | **0.0701** | **0.0856** | **0.0605** | **0.0553** | **0.0617** |

Table 11: P-values from McNemar's test comparing DynaDiag (Diag), SRigL, and PixelatedBFly with RigL at varying sparsities $s \in \{0.4, 0.5, 0.6, 0.8, 0.9\}$. Bold values indicate no significant difference (p-value $\geq 0.05$) from RigL.

| Model | Method | 40% | 50% | 60% | 80% | 90% |
|---|---|---|---|---|---|---|
| **GPT2-S** | RigL | - | - | - | - | - |
| | SRigL | 0.0456 | 0.0398 | 0.0321 | 0.0275 | 0.0213 |
| | PixelatedBFly | **0.0523** | 0.0471 | 0.0384 | 0.0332 | 0.0285 |
| | DynaDiag | 0.0183 | **0.0518** | **0.0675** | **0.0529** | 0.0401 |
| **GPT2-M** | RigL | - | - | - | - | - |
| | SRigL | 0.0395 | 0.0352 | 0.0287 | 0.0231 | 0.0178 |
| | PixelatedBFly | **0.0589** | 0.0427 | 0.0356 | 0.0304 | 0.0256 |
| | DynaDiag | **0.0518** | **0.0576** | **0.0547** | **0.0657** | 0.0136 |

Table 12: Top-1 accuracy of DynaDiag alongside baseline methods at varying sparsities. We bold results that are not significantly different from RigL based on paired asymptotic McNemar tests ($\alpha = 0.05$). Among all structured sparse training methods, DynaDiag achieves the highest accuracy on ViT-Small and MLPMixer architectures across CIFAR-10 and CIFAR-100 datasets.

| Models | Methods | Struc. | CIFAR10 | | | | | CIFAR100 | | | | |
|---|---|---|---|---|---|---|---|---|---|---|---|---|
| | | | 60% | 70% | 80% | 90% | 95% | 60% | 70% | 80% | 90% | 95% |
| | | | *dense accuracy = 85.64* | | | | | *dense accuracy = 66.98* | | | | |
| Mixer-S/16 | RigL | no | 86.44 | 86.47 | 86.74 | 85.85 | 84.65 | 67.44 | 67.97 | 67.54 | 66.52 | 64.2 |
| | SRigL | yes | **85.98** | **85.69** | 85.14 | 83.45 | 82.14 | **67.19** | **66.94** | **66.88** | 64.03 | 60.41 |
| | PixelatedBFly | yes | 85.39 | 85.25 | 84.69 | 82.41 | 81.09 | 66.45 | 65.74 | 64.36 | 63.57 | 59.56 |
| | DSB | yes | 83.14 | 83.45 | 82.14 | 80.21 | 80.12 | 65.14 | 64.74 | 64.26 | 62.44 | 58.14 |
| | DiagHeur. | yes | 82.69 | 83.01 | 81.11 | 79.56 | 78.65 | 64.54 | 64.21 | 63.47 | 62.3 | 60.08 |
| | **DynaDiag** | yes | **86.14** | **86.19** | **85.69** | **85.55** | **83.13** | **67.08** | **67.02** | **66.91** | **64.21** | **61.30** |
| | | | *dense accuracy: 89.67* | | | | | *dense accuracy: 66.61* | | | | |
| ViT-S/16 | RigL | no | 91.04 | 91.03 | 90.58 | 88.45 | 84.56 | 68.91 | 68.43 | 66.54 | 65.31 | 62.32 |
| | SRigL | yes | **90.67** | **90.51** | 89.32 | 86.88 | 81.54 | **68.51** | 67.85 | **66.21** | 64.81 | 61.11 |
| | PixelatedBFly | yes | 90.08 | 90.16 | 87.51 | 83.31 | 80.14 | 68.14 | **68.10** | **66.13** | 64.12 | 61.33 |
| | DSB | yes | 89.55 | 89.45 | 88.39 | 83.52 | 80.09 | 68.01 | 67.69 | 66.05 | 63.23 | 60.08 |
| | DiagHeur. | yes | 88.41 | 89.14 | 88.51 | 86.49 | 80.61 | 66.87 | 65.57 | 65.08 | 64.09 | 60.26 |
| | **DynaDiag** | yes | **90.55** | **90.63** | **89.43** | **87.09** | **82.76** | **68.41** | **68.12** | **66.31** | **65.06** | **61.43** |

Table 13: Perplexity of DynaDiag alongside baseline methods. We bold results that are not significantly different (calculated using paired asymptotic McNemar tests ($\alpha = 0.05$)) from the best performing method in the column (marked with a *). Among all the baselines, DynaDiag achieves the lower PPL (lower, the better) on the WikiText-103 dataset.

| Model | Method | 40% | 50% | 60% | 80% | 90% |
|---|---|---|---|---|---|---|
| | | *dense accuracy = 22.21* | | | | |
| GPT2-S | RigL | **22.34** | **22.80** | **23.79** | **29.87** | **53.76** |
| | SRigL | 22.74 | 23.19 | 25.09 | 31.08 | 62.55 |
| | PixelatedBFly | **22.50** | 23.25 | 25.98 | 34.89 | 66.44 |
| | Wanda | **22.14*** | **22.35*** | **23.48*** | **29.12*** | **53.07*** |
| | **DynaDiag** | 22.60 | **22.74** | **24.67** | **30.46** | 56.33 |

and ViT-Small in Tbl. 12, where DynaDiag performs as new state-of-the-art among structured DST methods, exceeding prior approaches at every sparsity level except one (60% on CIFAR10 with Mixer-S/16). The p-values under the asymptotic McNemar's test are reported in Tbl. 9 in Apdx. E.

We also observe that for CIFAR10 , the accuracy of both ViT-S/16 and Mixer-S/16 improves after sparsification compared to their dense counterparts. We hypothesize that this improvement stems from the fact that both models are **overparameterized**, meaning they possess more parameters than necessary to fit the training data. By sparsifying the models, we eliminate less important or redundant parameters, leading to a more efficient and generalizable model (Liu et al., 2019; Yang et al., 2023a).

### F.2. Comparison with Pruning Methods

Pruning LLMs (Sun et al., 2023; Kuznedelev et al., 2023) has been an effective way of reducing the size of the model while producing models with state-of-the-art inference accuracies at high sparsity levels. However, pruning requires dense training along with fine-tuning post pruning which is not the case with sparse-to-sparse training of models.

To see how DynaDiag performs as compared to pruning methods, we ran Wanda (Sun et al., 2023) with the GPT-2 small model and reported the results in Tbl. 13. As expected, Wanda, a pruning method, produces models with better perplexity than DST-based methods. However, Wanda's results are produced at a significantly higher computational cost (dense training + fine-tuning), whereas our method remains computationally efficient.

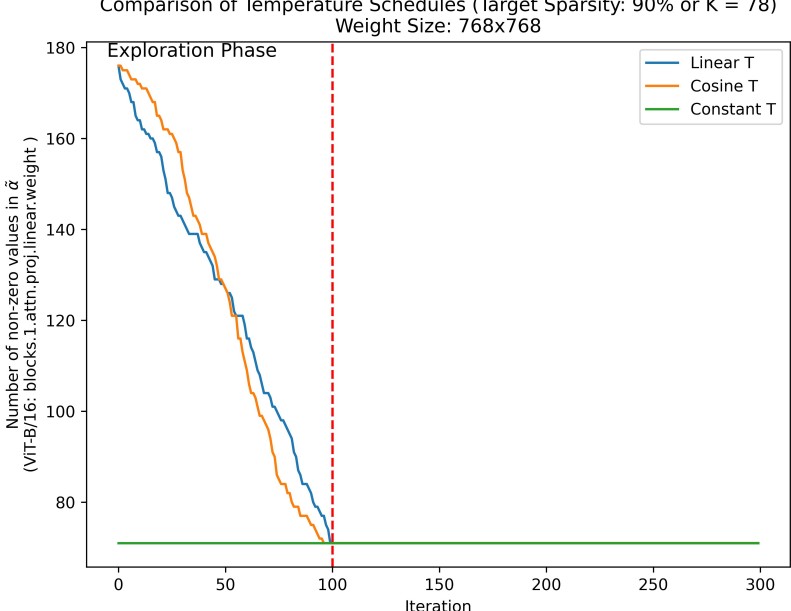

Figure 8: Comparing the three different temperature schedules which affect the amount of non-zeros present at a training step.

### F.3. Impact of Temperature Scheduling, Sparsity Scheduling, and Sparsity Distribution on Training

**Temperature Scheduling:** We study the evolution of non-zero entries in a representative weight matrix from ViT-B/16 under three temperature scheduling strategies—Linear, Cosine, and Constant—in Fig. 8. All approaches target 90% sparsity (i.e., retaining $K = 78$ values out of 768). We observe that both Linear and Cosine schedules gradually reduce the number of non-zeros during the early training iterations (exploration phase), allowing the model to adapt its sparse structure progressively. In contrast, the Constant schedule enforces the target sparsity from the beginning, leading to no exploration. This supports our finding that gradual temperature decay yields better performance by allowing more flexible structural learning early in training.

**Sparsity Scheduling:** We study the effect of different sparsity scheduling strategies—Constant, Linear, and Cosine—on the performance of ViT-B/16 trained on ImageNet-1K. These schedules control how the sparsity level evolves over training. As shown in Tbl. 15, the Cosine schedule consistently achieves the best accuracy across all sparsity levels, followed closely by Linear, while Constant performs significantly worse. This highlights the importance of a gradually decaying sparsity schedule for maintaining model accuracy under high sparsity regimes.

**Sparsity Distribution:** In our experiments, we allocate the sparsity budget, based on a layer's compute fraction (proportional to layer size) as proposed in Pixelated Butterfly (Dao et al., 2021)(Sec 3.3 and Appendix I.1). However, we also experiment with two other distributions: Uniform and ERK and the results are shown in Tbl. 14. We can see that computational fraction-based allocation yields better results, consistent with findings from Pixelated Butterfly.

Table 14: Impact of different sparsity distribution methods on ViT-B/16 performance on ImageNet-1K

| Method | 60% | 70% | 80% | 90% | 95% |
|---|---|---|---|---|---|
| Uniform | 77.64±0.0021 | 77.02±0.0013 | 76.73±0.0020 | 76.14±0.0015 | 69.31±0.0011 |
| Erdős-Rényi-Kernel (ERK) | 77.93±0.0018 | 77.43±0.0019 | 76.53±0.0014 | 76.21±0.0007 | 69.45±0.0016 |
| ComputeFraction (PBFly) | 78.29±0.0020 | 77.94±0.0017 | 77.62±0.0016 | 76.91±0.0016 | 69.54±0.0014 |

Table 15: Impact of different scheduling methods on ViT-B/16 performance on ImageNet-1K

| Method | 60% | 70% | 80% | 90% | 95% |
|---|---|---|---|---|---|
| Constant | 75.64±0.0013 | 74.82±0.0019 | 74.03±0.0012 | 72.74±0.0015 | 68.17±0.0016 |
| Linear | 77.93±0.0018 | 77.43±0.0019 | 76.53±0.0014 | 76.21±0.0007 | 69.45±0.0016 |
| Cosine | 78.29±0.0020 | 77.94±0.0017 | 77.62±0.0016 | 76.91±0.0016 | 69.54±0.0014 |

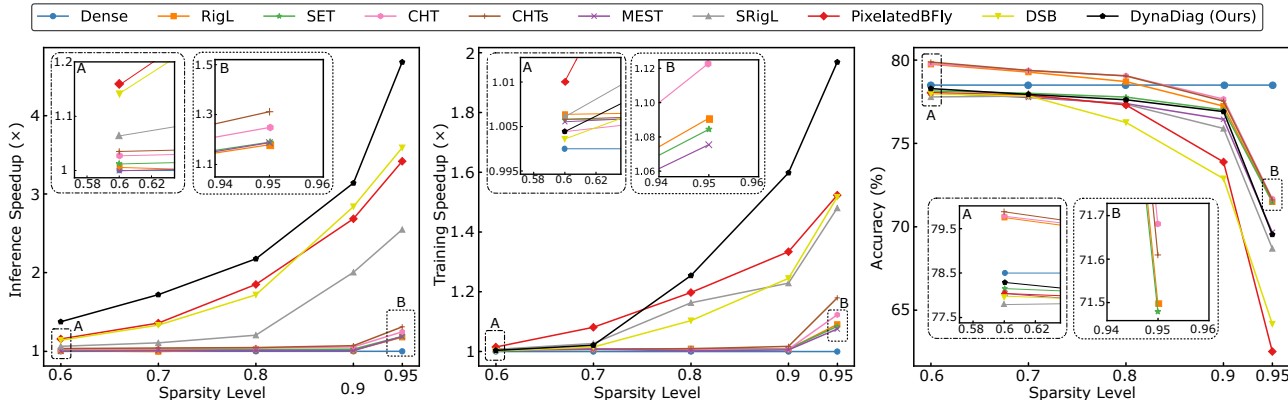

Figure 9: Comparing the inference speedup (left), training speedup (center), and Top-1 accuracy (right) of sparse training methods for ViT-B/16 model at varying levels of sparsity running ImageNet-1K. DynaDiag achieves the highest training and inference speedup while demonstrating superior accuracy to other structured sparsity patterns.

## G. Summary of Our Experiments

Fig. 9 shows the results of our experiments in a single figure. We can see that at 60% and 70% sparsity, DST accuracy performance is better than dense training, and that DynaDiag is the only structured method to keep its performance close to the unstructured ones that perform better than dense. DynaDiag shows a stable accuracy curve that remains high in comparison to the other structured methods. It is the only structured method to present a performance stability similar to unstructured methods across different levels of sparsity.

## H. Details of the Heuristics-Based Method

RigL is a dynamic sparsification technique that maintains a constant sparsity budget by repeatedly removing the lowest-magnitude weights (decay) and "growing" new connections. Similarly, in our DiagHeur method, we follow the same principle but focus on diagonal blocks: we prune the diagonals with the lowest magnitudes and randomly grow back the same number of diagonal connections, thereby preserving overall sparsity while continually redistributing capacity to potentially more valuable diagonals. We keep all the hyperparameters same as the original RigL (Evci et al., 2020) paper.

## I. Bipartite Small-World (BSW) and Bipartite Scale-Free (BSF)

**Bipartite Scale–Free (BSF).** The Bipartite Scale-Free (BSF) model of Zhang et al. (2024) adapts scale-free connectivity to bipartite graphs, making it ideal for dynamic sparse training. The procedure begins by sampling a standard Barabási–Albert (BA) graph (Barabási & Albert, 1999), whose node degrees follow a power-law distribution. Every edge that links two neurons in the same layer is then eliminated and re-attached to a neuron in the opposite layer. Crucially, each node keeps its original degree, so the resulting bipartite network retains the BA power-law exponent.

**Bipartite Small–World (BSW).** The Bipartite Small-World (BSW) model of Zhang et al. (2024) embeds small-world behaviour and strong clustering into a bipartite graph. It starts from a regular ring lattice, labeling the vertices with two distinct layer types. Every vertex connects to the same number of nearest neighbours that belong to the opposite layer, producing a highly clustered but not yet small—world structure. Analogous to the Watts–Strogatz construction (Watts

& Strogatz, 1998), the model then applies a rewiring rate $\beta$: a proportion $\beta$ of existing edges is randomly removed and re-attached elsewhere, introducing the shortcuts that create small-world characteristics.

### I.1. Diagonal Sparsity and Small World Networks

To test the small-worldness of networks trained with DynaDiag, we take a 90% sparse ViT-B/16 network trained on ImageNet-1K and calculate the small-world factor, $\sigma$, using the NetworkX library in Python. Tbl. 16 shows that all tested layers exhibit $\sigma \geq 1$, confirming that DynaDiag's structured sparsity indeed reflects small-world characteristics.

Table 16: Small-world factor, $\sigma$ of various layers in a DynaDiag trained ViT-B/16 on ImageNet-1K at 90% sparsity. $\sigma > 1$ indicates small world.

| Layer | $C_r$ | $L_r$ | $C$ | $L$ | $\sigma$ |
|---|---|---|---|---|---|
| blocks.0.attn.proj.linear.weight | 0.032 | 2.14 | 0.063 | 3.94 | 1.069 |
| blocks.1.mlp.fc1.linear.weight | 0.039 | 2.67 | 0.072 | 3.67 | 1.343 |
| blocks.2.mlp.fc2.linear.weight | 0.041 | 2.64 | 0.084 | 3.55 | 1.524 |
| blocks.3.attn.proj.linear.weight | 0.047 | 2.94 | 0.088 | 3.95 | 1.394 |
| blocks.4.mlp.fc1.linear.weight | 0.029 | 2.15 | 0.061 | 3.87 | 1.169 |
| blocks.5.mlp.fc2.linear.weight | 0.049 | 3.42 | 0.087 | 3.31 | 1.835 |
| blocks.6.attn.proj.linear.weight | 0.061 | 2.65 | 0.099 | 3.91 | 1.100 |
| blocks.7.mlp.fc1.linear.weight | 0.041 | 2.71 | 0.097 | 3.90 | 1.644 |
| blocks.8.mlp.fc2.linear.weigh | 0.032 | 2.83 | 0.066 | 3.88 | 1.504 |
| blocks.9.attn.proj.linear.weight | 0.047 | 2.23 | 0.081 | 3.77 | 1.019 |
| blocks.10.mlp.fc1.linear.weight | 0.054 | 3.06 | 0.096 | 4.13 | 1.317 |
| blocks.11.mlp.fc2.linear.weight | 0.051 | 3.11 | 0.097 | 3.97 | 1.490 |

