# OpenReview forum: "Dynamic Sparse Training of Diagonally Sparse Networks"
_ICML.cc/2025/Conference — ICML 2025 poster_

### Official Review · Reviewer_NGkq · 2025-03-09

**Overall Recommendation:** 3

**Summary:**

This paper introduces DynaDiag, a structured Dynamic Sparse Training (DST) method that enforces a diagonal sparsity pattern in neural networks. The method aims to overcome the inefficiencies of unstructured sparsity, which struggles to translate into hardware acceleration. The authors propose a custom CUDA kernel to optimize computations, making DynaDiag more hardware-friendly.

**Claims And Evidence:**

The paper is generally clear, and the claims are supported by empirical evaluations on vision and language models. The reported inference acceleration (3.13×) and training speedup (1.59×) on GPUs at 90% sparsity is backed by experimental results. However, I question the necessity of LoRA-FA fine-tuning in the context of the paper's efficiency claims, as it introduces unstructured sparsity, which may counteract the structured sparsity benefits.

**Essential References Not Discussed:**

I did not find any critical missing references that significantly impact the understanding of the paper’s contributions.

**Experimental Designs Or Analyses:**

No standard deviation for accuracy results. At extreme sparsity levels, instead of focusing solely on high sparsity, it might be more insightful to scale up the model first and then test at high sparsities while keeping the total parameter count comparable.

**Methods And Evaluation Criteria:**

The proposed methods and evaluation criteria are generally appropriate, as they include evaluations on vision and language tasks across different architectures. However, analyzing the applicability of DynaDiag to CNN-based models could strengthen its claims, as CNNs are still widely used in real-world applications.

**Other Comments Or Suggestions:**

### Some sections are confusing:
- In the beginning of the second paragraph, the description of randomly pruning weights is unclear and confusing.
- In Section 2.2, Paragraph 3, RigL removes weights based on magnitude, while regrowth is based on gradient-this should be clarified.
- In Section 2.2, Paragraph 4, Mocanu (2018) is an earlier work on DST, which should be acknowledged appropriately.
- Bold text in all tables should be double-checked, as it may be misleading or confusing.

**Other Strengths And Weaknesses:**

### Strengths

- This paper introduces a hardware-friendly approach that leverages the Block Compressed Sparse Row (BCSR) format to optimize GPU execution. While it successfully accelerates both inference and training, the full potential of sparsity speedup for training has yet to be fully realized.

- The introduction of diagonal sparsity, inspired by small-world networks, enables the method to scale effectively for large ViT models and GPT-2, demonstrating its robustness across different architectures and modalities.

### Weaknesses
- It would be useful to include a theoretical comparison explaining why the diagonal sparsity pattern works better under certain conditions. The paper lacks a clear, intuitive explanation for why it performs better than other structured sparsity patterns and why it achieves performance comparable to unstructured sparsity.
- Given that CNNs are still widely used in real-world applications, discussing the potential impact of DynaDiag on CNNs would strengthen the paper.
- The necessity of LoRA-FA for improving DST efficiency in this paper is unclear. Since it introduces unstructured sparsity after fine-tuning (as mentioned in Section 4.3.1, Paragraph 3), how does it affect inference speedup?
- How is the BCSR transformation performed, and how time-consuming is it? The effectiveness of this operation should be further analyzed, ideally with an ablation study comparing accuracy and computation time with and without the BCSR transformation. The lack of impact analysis weakens the discussion.
- The paper lacks an in-depth analysis of speedup at lower sparsity levels. In this case, what causes slower training but faster inference?
- Is it valuable to explore extreme sparsity levels if it only slightly outperforms RigL? How about scaling up the model first and then exploring extreme sparsity levels? Would it be more informative to maintain the same number of parameters under 90% sparsity for the original models and test both accuracy and computation time?

**Questions For Authors:**

This paper introduces a new structured sparsity approach that improves both accuracy and hardware speedup, paving the way for advancements in the sparse training community. Please consider the provided comments, and these questions are open for discussion.

**Relation To Broader Scientific Literature:**

The paper is well-situated within the broader field of Dynamic Sparse Training (DST) and relates well to prior work. However, some claims could be further refined with more details in the suggestions section.

**Theoretical Claims:**

The paper presents a relatively good theoretical foundation, particularly in its proof of transposability for diagonal sparse matrices and Top-K selection mechanism. However, the justification for diagonal sparsity is weak. While inspired by small-world networks, the paper lacks an intuitive explanation for why diagonal sparsity performs better than other structured sparsity methods (e.g., block sparsity, N:M sparsity). A comparative theoretical analysis would be beneficial.

---

> ### Author Rebuttal · Authors · 2025-04-01
>
> We thank the reviewer for their comments and feedback, especially about grounding our work in theoretical foundations. We address the questions and comments below:
>
> ## LoRA-FA Finetuning
>
> We used LoRA-FA (Sec 4.3.1) primarily to interpret the performance gap between RigL and DynaDiag by introducing limited unstructured sparsity. While we acknowledge that LoRA-FA introduces additional computational overhead at inference, this was purely an exploratory experiment rather than a core component of our method. To avoid confusion, we will rename the section to “Interpreting the gap between unstructured sparsity and DynaDiag” in the final version.
>
> ## DynaDiag on CNNs
>
> We agree with the reviewer, and we are working towards extending DynaDiag for CNNs. The current implementation of DynaDiag uses one $alpha$ vector for each linear layer. This approach is not scalable to a CNN directly as there are many weight matrices (filters) for each layer, and each matrix can have diagonals in different positions, resulting in an $alpha$ vector per filter and increasing the memory overhead.
>
> ## Exploring Extreme Sparsity
>
> We agree with the reviewer that it will be beneficial to compare DynaDiag’s performance with unstructured sparsity by scaling the models up. For such a study, we ideally need models that posses more than one diagonal when sparsity reaches levels such as 99.999\%, where RigL has been shown to collapse [[1]](https://openreview.net/pdf?id=szRHR9XGrY). Our currently chosen models do not possess such properties, and hence, we welcom any suggestions from the reviewer on which models would be suitable for such a study.
>
> ## Reporting Standard Deviations
>
> We do not report std dev values at this stage as we find that our training is relatively stable for vision tasks and language tasks, which has previously been the case [[2]](https://openreview.net/pdf?id=MWZWUyfFHC). However, we plan to add more elaborate results to the final version of our manuscript.
> We do, however, report the p-values from our statistical significance tests for all the experiments in Table 8, Table 9, and Table 10 in Appendix A.4.
>
> ## Theoretical Explanation of DynaDiag’s Performance
>
> We submit a theoretical justification of why diagonal sparsity works better than other structured sparsity patterns.
>
> **Theoretical Explanation:** We request the reviewer to check out our [theoretical explanation for DynaDiag’s performance](https://figshare.com/s/06ea8e6cf2c4b86f38f0).
>
> **Conclusion:** Briefly, we show that diagonal sparsity offers a full input-output coverage i.e. each row and column contains at least one non-zero entry in each weight matrix, even with a single diagonal). This:
>
> 1) Preserves the rank of the matrix, maintaining the same expressivity as that of the dense matrix.
> 2) Ensures global input-output connectivity and coverage, avoiding dimension collapse.
> 3) Preserves theoretical guarantees of universal approximation capability.
>
> ## Diagonal-to-BCSR Transformation
>
> We request the reviewer to kindly check our [response to the Reviewer vAzp](https://openreview.net/forum?id=bAUVnNc0Ky&noteId=HoMgMLdsaV) under the section “BCSR Transformation and Contribution” and “Performance Impact of Diags to BCSR conversion (Q4)”, where we discuss how the conversion takes place and its impact on the performance.
>
> ## Speedup at Lower Sparsities
>
> At lower sparsity levels, we have a larger (but still less than dense) number of diagonals in all weight matrices. We see an increase in training time as compared to high sparsity due to the following overhead:
>
> Diag to BCSR conversion: We find that the time taken to convert a diagonal matrix to BCSR using our method is proportional to the number of diagonals in the matrix. For example, for a projection matrix (ViT-B/16) of size 768x768 in an attention block, the total conversion time varies as 4.35 us, 5.23 us, 7.78 us, and 10.53 us at sparsities 90%, 80%, 60%, and 50% respectively.
>
> DynaDiag can still accelerate inference at lower sparsities as the conversion of diagonal matrices to BCSR takes place offline and is done just once at the start of inference.
>
> ## Writing Edits
>
> We appreciate the reviewer’s feedback regarding clarity. We commit to improving:
>
> 1) The description of random pruning in the introduction.
>
> 2) Clarifying RigL’s growth strategy in Sec 2.2, Para 3.
>
> Proper acknowledgment of SET approach in Sec 2.2, Para 4.
>
> Verifying and updating bolded values in tables to clearly reflect best-performing methods.
>
> These edits will be included in the camera-ready version of our manuscript. We thank the reviewer again for their valuable input.

---

> > ### Comment · Reviewer_NGkq · 2025-04-03
> >
> > I greatly appreciate that the authors have completed numerous experiments within a limited time, and most of my initial concerns have been addressed. I would raise my score to 3. Nonetheless, there remain some concerns and suggestions:
> >
> > 1. Regarding lower sparsity levels, the authors note that a larger (albeit still less than dense) number of diagonals in weight matrices results in only marginal speed improvements. I recommend including visualizations or other analytical approaches to clearly demonstrate how the number of diagonals correlates with speedup across varying sparsity levels.
> >
> > 2. Model scaling, Figure 6 indicates a substantial performance decline at extremely high sparsity levels, potentially restricting practical applicability. Therefore, I suggest exploring the scaling of models under these high sparsity conditions, as in [1], as this could reveal more compelling insights.
> >
> > [1] Sparse-IFT: Sparse iso-FLOP transformations for maximizing training efficiency[J]. ICML, 2024.

---

> > > ### Author Response · Authors · 2025-04-08
> > >
> > > We thank the reviewer for raising their score and address their concerns below:
> > >
> > > ## Performance at Extreme Sparsity
> > >
> > > Thank you for this insightful suggestion. As shown in Figure 6 of the paper, all methods—including unstructured approaches (Jie et al., 2024)—exhibit a noticeable performance drop at extreme sparsity levels. While DynaDiag rivals RigL at these extremes and outperforms other structured patterns, such a decline is unsurprising at 99.9% sparsity, where some layers are reduced to just three diagonals.
> > >
> > > We are actively exploring how scaling models width-wise, in line with references like Sparse-IFT (Thangarasa et al., 2023), might mitigate this effect under ultra-high sparsity. However, these new results will not be ready before the rebuttal deadline. We appreciate your feedback and look forward to incorporating these insights in future work.
> > >
> > > ## Speedup at varying sparsity level
> > >
> > > Thank you for this constructive suggestion. We already included performance measurements for DynaDiag at 60–90% sparsity in Figure 4 of our paper.
> > >
> > > To address your request for even lower sparsity insights, we performed an additional experiment on an NVIDIA A100 GPU with a 768×768 matrix-matrix multiplication (matching the size of the blocks.I.attn.proj.linear.weight layer in ViT where I is the block count) to isolate the impact of number of diagonals on potential speedup. Each configuration was run 100 times, and we averaged the total time of converting diagonals to BCSR plus the subsequent BCSR computation.
> > >
> > > **Result:** The [plot with speedups can be accessed here](https://figshare.com/s/7f2de1bed877c35917a5).
> > >
> > > **Conclusion:** As expected, below ~50% sparsity, speed gains taper off, and below ~20% sparsity, we see some slowdown—yet this remains more favorable than comparable block-sparse approaches (Takuma et al., 2021).
> > >
> > > We hope these results clarify how the number of diagonals (i.e., sparsity levels) affects speedup, and we appreciate your consideration of our updated analyses.
> > >
> > >
> > > ## References
> > >
> > > Thangarasa, Vithursan, et al. "Sparse-IFT: Sparse iso-FLOP transformations for maximizing training efficiency." arXiv preprint arXiv:2303.11525 (2023).
> > >
> > > Ji, Jie, et al. "Advancing dynamic sparse training by exploring optimization opportunities." Forty-First International Conference on Machine Learning. 2024.
> > >
> > > Yamaguchi, Takuma, and Federico Busato. "Accelerating matrix multiplication with block sparse format and nvidia tensor cores." NVIDIA Developer Technical Blog, https://developer. nvidia. com/blog/accelerating-matrixmultiplication-with-block-sparse-format-and-nvidia-tensor-cores (2021).

---

### Official Review · Reviewer_vAzp · 2025-03-14

**Overall Recommendation:** 4

**Summary:**

This paper introduces DynaDiag, a novel structured Dynamic Sparse Training (DST) method based on the composition of diagonal matrices. This parametrization is then transformed into the BCSR format to enable actual GPU acceleration. The proposed model is evaluated by training from scratch vision VIT and MLP-Mixer models, as well as the GPT-2 model, closing the gap between the unstructured and structured DST algorithms.

**Claims And Evidence:**

The claims in the paper are mostly clear, supported either by empirical analysis or previous research. I do have, however, one issue with the contributions. In particular, with the last point. The paper presents the conversion of matrices to BCSR format as a contribution, but it remains unclear how novel this conversion method is. Does it simply apply existing techniques from Section 3.3, or does it introduce new fundamental ideas?

**Essential References Not Discussed:**

I do not indicate any missing essential references at this point

**Experimental Designs Or Analyses:**

The experimental designs and analyses seem sound (Table 1, Table 2, Figure 4, Figure 1, Table 8). However, in terms of replicating the experiments, the Appendix lists the used hyperparameters in tables, but the section responsible for their summary (A.2.2) lacks reference to those tables. Furthermore, it would be beneficial to explain why such configurations were used (are they an effect of grid search, or maybe they were adapted from literature etc.)

**Methods And Evaluation Criteria:**

The paper evaluates DynaDiag primarily on ViT and MLP-Mixer architectures using vision datasets, with a single experiment on GPT-2 for text. The selected baselines include other common structured sparse-to-sparse methods and RigL as the sole representative of unstructured Dynamic Sparse Training (DST). While RigL is a strong baseline—often matching dense model performance in sparse training for vision tasks—expanding the evaluation to additional LLM architectures would strengthen the analysis.
I recognize that pretraining larger LLMs may be computationally prohibitive, but exploring sparse-to-sparse finetuning (e.g., using LLaMA) could offer valuable insights. This would not only increase model diversity but also allow for comparisons with widely used pruning-based methods like Wanda and SparseGPT. Additionally, investigating the impact of dataset size could make the paper more relevant to the broader research community and enhance its impact.

**Other Comments Or Suggestions:**

-

**Other Strengths And Weaknesses:**

Strengths:
-  The proposed method is built on an elegant idea of parametrizing linear projection matrices as a composition of diagonal matrices.
-  It helps bridge the gap between unstructured Dynamic Sparse Training (e.g., RigL) and structured approaches while also reducing computational time, as measured in terms of wall-clock time.
-  The paper is well-structured and easy to follow. The core idea is simple yet elegant and shows great promise both in empirical evaluations and in its potential for real-world speedups.

Weaknesses:
- I have discussed the weaknesses in the corresponding paragraphs above.

**Questions For Authors:**

1. What is the precise contribution regarding the transformation to the BCSR format? The authors present this as a key contribution, but Section 3.3 leaves some ambiguity. Are they merely applying existing methods, or does their approach introduce novel aspects? Clarifying this is essential for assessing whether the paper fully delivers on its claims.
2. Could the evaluation be extended to sparse-to-sparse finetuning? (See Methods and Evaluation Criteria.) Including this aspect could  enhance my assessment of the experimental results.

Clarification Questions (to ensure a proper understanding of the paper):

  3. The paper states, "We derive the per-layer sparsity budget from the global sparsity budget." How exactly is per-layer sparsity distributed, and how do different distribution strategies impact DynaDiag’s performance?
  4. Additionally, could the BCSR conversion influence overall performance? How would DynaDiag perform if implemented naïvely using simple masking instead?

Minor (curiosity-driven) Questions (unlikely to impact my overall score but still of interest):

   5. What effect do temperature and alpha initialization have on DynaDiag’s final results?

**Relation To Broader Scientific Literature:**

Dynamic Sparse Training (DST) is a well-established field focused on developing algorithms for training models that are sparse from initialization. The term “dynamic” refers to methods that adaptively modify the connectivity or sparse structure throughout training.

The core contribution of this paper is the introduction of a new structured DST algorithm. While prior structured DST approaches, such as SparseRigL, have been proposed and are discussed by the authors, I am not aware of any previous attempts that leverage the composition of diagonal matrices as presented here.

More broadly, in the context of sparsity in transformer-based models, this work is also relevant to pruning-at-initialization methods. Consequently, I believe the paper would benefit from a comparison with pruning approaches such as Wanda or SparseGPT to provide additional context and insights.

**Theoretical Claims:**

The paper does not include theorems or proofs, the claims are evaluated via empirical experimentation.

---

> ### Author Rebuttal · Authors · 2025-04-01
>
> We thank the reviewer for their feedback and positive comments. We are glad to hear that the
> reviewer appreciates the novelty of our work. We address the questions and comments below:
>
> ## BCSR Transformation and Contribution (Q1)
> For acceleration on GPUs, a diagonal matrix is converted to blocks (represented in BCSR) by calculating a row transformation $P$.
>
> **Contribution:** We propose a heuristic-based method to convert diagonal matrices to block matrices which is built on top of SMAT[[1]](https://arxiv.org/pdf/2408.11551), which proposes a method to calculate $P$ for the general case of unstructured matrices.
>
>  Where SMAT uses Jaccard similarity between the rows to block them efficiently (see Section IV C of SMAT), we modify Jaccard’s similarity (as mentioned in detail in Appendix A.3) with a diagonal proximity term to prioritize diagonals close to the main diagonals, which maximizes the total number of dense blocks. Our heuristics take advantage of the fact that each row in the matrix has the same number of nonzeros which help with load balancing when running on the GPU.
>
> ## Performance Impact of Diags to BCSR conversion (Q4)
> Converting a diagonal matrix $W_{diag}$ to a blocked matrix (in BCSR) $W_{BCSR}$ is done by applying row permutation to $W_{diag}$ given as:
>
>  $W_{BCSR} = P_{perm}W_{diag}$
>
> **Matrix-Vector equivalence in diag and BCSR Format:** To ensure that matrix-vector multiplication remains equivalent between the two formats, an inverse of $P_{perm}$ (which is its transpose) is applied to the incoming input $x$. We show the equivalence below:
>
> We can write:
>
> $Output_{diag} = W_{diag}X$
>
> $Output_{diag} = W_{BCSR}P_{perm}^{-1}X$
>
> $Output_{diag} = W_{BCSR}(P_{perm}^{-1}X)$
>
> Therefore, the two methods result in the same output.
>
> **Experiment:** With empirical experiments on ViT-B/16 at 90% sparsity with ImageNet-1K, we verify that there is no significant accuracy difference between direct diagonal computation and the BCSR-based approach.
>
> **Results:** Please refer to [Table 16 here](https://figshare.com/s/b6386a7f35eca3ebcfeb).
>
> **Conclusion:** We see no significant difference in the accuracies of the two methods, proving their equivalence. However, the training time is significantly improved using our custom BCSR kernel.
>
> ## Comparison with Sparse-to-Sparse finetuning methods (Q2)
> We agree with the reviewers' suggestion to apply our training method to LLaMA models for comparison with SOTA pruning methods such as Wanda and SparseGPT. Doing so will require us to train a LLaMA model from scratch, and hence, **we propose to share those results in our final manuscript.**
>
> However, we ran Wanda (SparseGPT did not work out of the box with GPT-2) with the GPT-2 model and reported the results.
>
> **Results:** Please refer to [Table 15 here](https://figshare.com/s/b6386a7f35eca3ebcfeb).
>
> **Conclusion:** As expected, Wanda, a pruning method, produces models with better perplexity than DST-based methods. However, the performance difference is not significant compared to DST methods (as evident from McNemar’s test results). However, Wanda's results are produced at a significantly higher computational cost (dense training + fine-tuning), whereas our method remains computationally efficient.
>
> ## Per Layer Sparsity (Q3)
> Please refer to our [response to Reviewer XE14](https://openreview.net/forum?id=bAUVnNc0Ky&noteId=K2Jst84n3i), under the section titled: “Per Layer Sparsity”.
>
> ## Justification For The Choice Of Training Configurations
> We thank the reviewer for pointing out the missing justification for the choice of training configurations in Tables 3, 4, 5, 6, and 7. All the configurations are obtained from the literature, and here are the references:
>
> 1) Tbl 3 [[2]](https://shorturl.at/QFzJw)
> 2) Tbl 4 [[3]](https://shorturl.at/bc3hz)
> 3) Tbl 5 and Tbl 7 [[4]](https://shorturl.at/4Hzxp)
> 4) Tbl 6 [[5]](https://shorturl.at/4qmmR)
>
> ## Temperature and Alpha Initialization (Q5)
>
> ### Temperature:
>
> We employ a temperature-based softmax for differentiable TopK approximation. Initially set high (T=1.0), the temperature gradually anneals to T=0.01, focusing the search from broad exploration to a sharp selection of diagonals. This approach balances exploration and computational overhead effectively, guided by prior methods such as Sander et al [[6]](https://dl.acm.org/doi/10.5555/3618408.3619649).
>
> ### Alpha Initialization:
>
> Currently, all $\alpha$s are initialized equally, assuming initial equal importance across diagonals. For future exploration, we are working with two alternative initialization methods:
>
> **Distance-Based Initialization:** Higher initial values for diagonals near the main diagonal.
>
> **Data-Driven Initialization:** Input data-driven initial importance reflecting intrinsic data structure or correlations.
>
> These strategies may enhance convergence speed and overall performance.
>
> We appreciate the reviewer’s valuable suggestions, which can considerably improve the clarity and strength of our paper.

---

> > ### Comment · Reviewer_vAzp · 2025-04-05
> >
> > Thank you for the clarifications. I appreciate both the quality and the number of additional comparison experiments conducted for the rebuttal.
> >
> > One observation I have, however, is that the performance of DynaDiag appears to degrade more significantly with increasing sparsity compared to DST or Wanda methods (as seen in Tables 11 and 15). Do the authors attribute this to the structured vs unstructured sparsity setups?
> >
> > Overall, assuming the additional results on DST and Wanda are incorporated into the main text, I am willing to raise my score.

---

> > > ### Author Response · Authors · 2025-04-07
> > >
> > > We thank the reviewer for their support of our work. We address their query below.
> > >
> > > ## DynaDiag at Higher Sparsity
> > >
> > > DynaDiag consistently outperforms other structured sparsity methods, reflecting both the expressiveness of its pattern and the strength of our training strategy. However, at extreme sparsities (>95%), it does experience a slightly higher degradation (e.g., ~2.14% at 95% sparsity vs. CHT in Table 11) compared to unstructured methods. We do attribute this to the inherently higher flexibility of unstructured sparsity.
> > >
> > > Nonetheless, we believe insights from scaling sparse networks (as mentioned by the reviewer NGkq) akin to Sparse-IFT (Thangarasa et al., 2023) could help close the gap between DynaDiag and unstructured approaches such as DST and Wanda at these extreme sparsity levels.
> > >
> > > ## Reference
> > >
> > > Thangarasa, Vithursan, et al. "Sparse-IFT: Sparse iso-FLOP transformations for maximizing training efficiency." arXiv preprint arXiv:2303.11525 (2023).

---

### Official Review · Reviewer_XE14 · 2025-03-14

**Overall Recommendation:** 5

**Summary:**

The paper introduces DynaDiag, a novel structured Dynamic Sparse Training (DST) method that enforces diagonal sparsity to improve both computational efficiency and model accuracy. Unlike unstructured DST methods, which often fail to achieve hardware acceleration despite high sparsity ratios, DynaDiag maintains structured sparsity throughout training, enabling significant speedup in both training and inference. The key idea is to enforce a diagonal sparsity pattern that remains structured under transposition, making it well-suited for efficient GPU execution. To optimize learning, DynaDiag employs a dynamic TopK-based selection mechanism to adaptively update the most critical diagonals, allowing the model to learn the optimal sparse connectivity during training. Additionally, the authors introduce a custom CUDA kernel that utilizes Block Compressed Sparse Row (BCSR) format, ensuring fast execution for both forward and backward passes. Empirically, DynaDiag achieves a 3.13× inference speedup and a 1.59× training speedup compared to dense models, while matching or outperforming existing structured sparse training methods. The method is evaluated on Vision Transformers (ViTs), CNNs, and GPT-2 models across ImageNet, CIFAR-10, CIFAR-100, and WikiText-103, demonstrating strong generalization even at extreme sparsities (up to 99.99%). The results indicate that DynaDiag is a viable alternative to unstructured DST, offering comparable accuracy while significantly improving computational efficiency. The study highlights the limitations of existing sparse training approaches and presents DynaDiag as an effective way to achieve both hardware acceleration and model performance, making it highly promising for scaling sparse deep learning models in practical applications. By bridging the gap between sparse training and hardware efficiency, DynaDiag paves the way for more efficient deep learning architectures, especially in resource-constrained environments.

**Claims And Evidence:**

About the claim that “Empirical evaluations on diverse neural architectures demonstrate that our method maintains accuracy on par with unstructured counterparts while benefiting from tangible computational gains.” I think it should be adjusted and reduced because there are many new evidences in literature in which unstructured methods such as CHT, CHTs and CHTss can perform better than RigL, GMP and even better than fully connected architectures (see the section below on essential references not discussed).

I agree of course about these second part of the claim “while benefiting from tangible computational gains.”
In my opinion the current version of the paper do not report solid evidence that supports the claim because it neglects to compare with the most recent methods in unstructured and structured dynamic sparse training (see the section below on essential references not discussed). In addition, also the related work and the Figure 1 and 2 and the other section of the article should be adjusted reporting appropriate claims on the most recent methods in the literature.

**Essential References Not Discussed:**

Mocanu, D. C.,et al. Scalable training of artificial neural networks with adaptive sparse connectivity inspired by network science. Nature Communications (2018), 9, Article number: 2383.

Yuan, G., et al. "MEST: Accurate and fast memory-economic sparse training framework on the edge." Advances in Neural Information Processing Systems 34 (2021): 14476-14489.

Zhang, Y. et al. Epitopological learning and Cannistraci-Hebb network shape intelligence brain-inspired theory for ultra-sparse advantage in deep learning. Proceedings of the Twelfth International Conference on Learning Representations (ICLR) 2024.

Wu, B., et al. Dynamic Sparse Training versus Dense Training: The Unexpected Winner in Image Corruption Robustness. Proceedings of the Twelfth International Conference on Learning Representations (ICLR) 2025.

Zhang, Y., et al. "Brain-inspired sparse training enables Transformers and LLMs to perform as fully connected." arXiv preprint arXiv:2501.19107 (2025)

**Experimental Designs Or Analyses:**

Yes, I checked, and I believe the manuscript should be improved. I recommend to test and discuss result including MEST, CHTs, CHTss  (see the section below on essential references not discussed).

**Methods And Evaluation Criteria:**

Yes, they make sense but the number of tested methods are few. I recommend discussing and add SET, MEST, CHTs, CHTss (see the section below on essential references not discussed).

**Other Comments Or Suggestions:**

Please see the question section

**Other Strengths And Weaknesses:**

Strengths
This article introduces the diagonal sparse training which is novel in this field. Very nice idea.
The proposed methods outperform the other structured sparsity sparse training methods in Comprehensive Benchmarking on Vision and Language Tasks

Weakness:
The study section on related work is incomplete and far from reporting the most recent advancements in the field.
The study needs to compare with SOTA unstructured methods to support their claims and offers an updated comparison of their method.
A lot of the training details of this article are unclear and the work necessity a robust revision to match the standard of reproducibility

**Questions For Authors:**

I kindly ask to the authors whether they can revise the study:

1. balancing their claims under the light of my comments above and the new literature I provided.
2. rewrite all the articles sections, and in particular the related work section, taking in consideration my comments and the new literature I provided.
In particular:
 Fig1 should modified including if possible other methods such as SET, MEST, CHTs, CHTss.
Fig. 2 should be modified including other sparsity patterns such as bipartite small-world (BSW, which for beta=1 includes the random pattern) and bipartite scale-free (BSF) modelling or with methods such as CSTI which create a regular diagonal pattern that is input data based.
3. The authors should offer a quantitative proof that their topologies are really small-world using the small word measures. The claim is not really theoretically well posed from the network science standpoint, it necessity evidences by small-worldness measures to offer evidence that their sparsity patter is characterized by the small-world phenomenon. To me it seems that their sparsity pattern is more inspired by a regular highly clustered pattern which is indeed very similar to CSTI (see Zhang et al ICLR2014 in the section Essential References Not Discussed) or BSW with beta=0 (which in reality is not small-world because the Watts-Strogatz small-world model for beta=0 is a regular clustered structure for which the small-world effect vanishes). But I might be wrong and for this I kindly ask a quantification of small-worldness.
4. insert in the computational experiments, when it is possible , the comparison with new methods such as SET, MEST, CHTs, CHTss.
5. How the alpha initialized? If alpha is initialized equally, then it will start from a fully connected network. Then how is the learned density decay curve?
6. I'm not sure how the unstructured sparse training method RigL is tested. As the article said, "We use CUSPARSE, as RigL lacks exploitable structure." Which kind of sparse format is the author using? Can you provide the single module running time of the proposed diagonal sparse training/inference and the other structures? As the density changes, the running time of RigL doesn't change. However, as my experience, if using the torch.sparse module, the running time will change significantly based on sparsity. However as shown in the article, the running time doesn't change from 60-90% sparsity.
7. Could the authors compare their method with CHTs [1], which has been shown to outperform RigL in most cases, as a baseline for unstructured sparse training?
8. In Figure 3a, what are the arguments of TopK(α, t, s)? Specifically, what do t and s represent?
9. The per-layer sparsity budget ρj is derived from the global sparsity budget ρglobal, and the number of diagonals Kj for each layer is determined based on ρj. Does the paper adopt a non-uniform sparsity distribution across layers? And what does the final sparsity distribution look like.
10. In Formula (5), the authors use min(max(oi, 0), 1). However, shouldn't oi always be ≥ 0? Also, what does k represent in the formula, and how is it tuned?
11. The authors mention that to encourage sparsity in α, they employ an ℓ1 regularization term. Could they elaborate on how this is implemented and its impact on training?
12. The authors apply a cosine-annealing schedule to adjust T during training, starting from a high value for a smoother TopK selection. Have they tested alternative scheduling methods?
13. Many training details is missing which makes the article hard to reproduce. If the article is accepted, do the authors plan to release the code?

**Relation To Broader Scientific Literature:**

The idea and method of this article is nice and solid, and I believe that if the authors address my concerns this article would be interesting for all the community.

However, considering the current version of the paper, I think that the relation to Broader scientific literature should be significantly improved. The authors are not aware of relevant new finding in unstructured/structured dynamic sparse training and that there are today also methods such as CHT, CHTs and CHTss that can perform better than fully connected networks. In addition, the authors do not discuss that the network topology initialization with network science modelling such as bipartite small-world and bipartite scale-free modelling or with method such as CSTI that are input data based in Figure. 2.
The study overclaims the relevance of their results with respect unstructured methods not comparing with current SOTA.
I invite the authors to largely revise the study to put in the appropriate and balanced context, because I believe this study can be of value for the community.

**Theoretical Claims:**

I have checked the proof of this article and it is correct

---

> ### Author Rebuttal · Authors · 2025-04-01
>
> We thank the reviewer for their insightful and constructive feedback, which has significantly helped enhance our work. Below, we concisely address each comment:
> ## Comparison with SET, MEST, CHT, CHTs Methods (Response to Q1, Q4, Q7)
> Due to time constraints, we chose the above four of the five methods suggested by the reviewer and carried out experiments on ViT-B/16 with ImageNet-1K.
>
> **Results:** Please refer to [Table 11 here](https://figshare.com/s/b6386a7f35eca3ebcfeb) for performance numbers.  We also [updated Figure 1](https://figshare.com/s/3c8b904acdce99263d30) to incorporate the methods suggested by the reviewer.
>
> **Conclusion:** CHT and CHTs outperform RigL at most sparsities. We calculate the statistical significance (using McNemar’s test) of the performance at various sparsities, and find that all the models in bold can be considered equivalent for that sparsity level. DynaDiag’s performance is also equivalent to that of the best model at that sparsit.; Thus, our structured sparsity performs comparable to current SOTA unstructured methods.
>
> ## Small-Worldness (Response to Q3)
> We agree with the reviewer, and we take a ViT-B/16 trained with DynaDiag and calculate the small-world factor, $\sigma$ using NetworkX[[1]](https://shorturl.at/7qS5A) library in Python.
>
>  **Results:** Please refer to the results for various layers in [Table 12 here](https://figshare.com/s/b6386a7f35eca3ebcfeb)
>
> **Conclusion:** All tested layers exhibit $\sigma \geq 1$, confirming that DynaDiag's structured sparsity indeed reflects small-world characteristics.
>
> ## Alpha Initialization and Other Scheduling Strategies (Response to Q5 & Q12)
> All $\alpha$s are initialized equally to start with, which are then passed as input to our $TopK$ function. The output of the TopK function is a sparse vector $\tilde{\alpha_i}$, which is used to obtain a diagonally sparse weight matrix for a layer, given as:
> $W_{K} = \sum_{j=1}^K \tilde{\alpha}_j \,P_j \,\mathrm{diag}(V_j)$
>
> **Results:**
>
> We obtain a [density decay curve ](https://figshare.com/s/2b5d7955c1c92fb72c42).
>
> We also show how different scheduling methods affect the performance of ViT-B/16 [Table 14](https://figshare.com/s/b6386a7f35eca3ebcfeb).
>
> The duration of the exploration phase is a hyper-parameter, and we choose its value as $\frac{1}{3}\times epochs$.  The initial and final temperature value is set as described by Sander et al [[2]](https://shorturl.at/eOKWy).
>
> ## Training Time of RigL (Response to Q6)
> For training unstructured sparse networks, we use torch.sparse for doing sparse matrix-matrix operations. We report the wall clock time for both training and inference.
> For Fig 1 and Fig 4a, we use the inference time with a batch size of 1 to obtain per sample inference time.Although no substantial speedup was observed from 60% to 90% sparsity, consistent with SRigL[[3]](https://shorturl.at/HX30S) and SparseRT[[4]](https://shorturl.at/baywp), noticeable improvements appear at 95% sparsity.
>
> **Results:** We report the inference and training time in [Tables 17 and 18 here](https://figshare.com/s/b6386a7f35eca3ebcfeb), respectively.
>
> We welcome reviewer suggestions for enhancing our unstructured timing baseline.
>
> ## Per Layer Sparsity (Response to Q9)
> We allocate the sparsity budget, based on a layers compute fraction (proportional to layer size) as proposed in PixelatedBFly[[1]](https://shorturl.at/Gblym) (Sec 3.3 and Appendix I.1).
>
> **Experiment:** We also experiment with two other distributions: Uniform and ERK.
>
> **Results:** We tabulate the results in [Table 13 here](https://figshare.com/s/b6386a7f35eca3ebcfeb).
>
> **Conclusion:** Computational fraction-based allocation yields better results, consistent with findings from PixelatedBFly.
>
> ## Arguments of $TopK(\alpha, s, t)$ (Response to Q8)
> The TopK function arguments are the importance vector ($\alpha$), layer sparsity ($s$), and
> temperature ($t$), controlling the softness/hardness of the selection.
>
> ## Clarification on Equation (5) and L1 regularization of $\alpha$ (Response to Q10 & Q11)
> We updated the equation considering the reviewer’s comments:
>
> $\tilde{\alpha_i} = \min \Bigl(k \cdot
>    \frac{\exp\bigl(\tfrac{\alpha_i}{T}\bigr)}{\sum_{j=1}^n \exp\bigl(\tfrac{\alpha_i}{T}\bigr)}
> , 1\Bigr)$
>
> K is the total number of diagonals per layer, determined from per-layer sparsity (footnote 1 on page 3).
>
> We apply L1 regularization on alphas to encourage sparsity and we add the L1 loss to the overall loss function as:
> 	$\text{Loss} = \text{crossEntropyLoss} + \lambda \sum_{l=1}^{L} \|\alpha^{(l)}\|_1$
> The regularization coefficient $\lambda$ is set at 0.2 following a hyperparameter search, and we
> observe no noticeable impact on training speed.
>
> ## Do we plan to release the code? (Response to Q13)
> Yes. We also plan to provide a step-by-step guideguide for reproducibility.
>
> We sincerely hope these adjustments fully address the reviewer’s feedback and request reconsideration of our work based on these substantial enhancements.

---

> > ### Comment · Reviewer_XE14 · 2025-04-02
> >
> > 1. The Authors made a great effort to address my questions, for this reason I increase my score to from 2 to 3.
> >
> > 2. The Authors forgot to address my Q2:
> >
> > << rewrite all the articles sections, and in particular the related work section, taking in consideration my comments and the new literature I provided. In particular: Fig1 should modified including if possible other methods such as SET, MEST, CHTs, CHTss. Fig. 2 should be modified including other sparsity patterns such as bipartite small-world (BSW, which for beta=1 includes the random pattern) and bipartite scale-free (BSF) modelling or with methods such as CSTI which create a regular diagonal pattern that is input data based.>>
> >
> > If the Authors address also this last question, I will increase my score to 4  or above.
> >
> > thanks for this excellent job.
> >
> > **Reply to Authors rebuttal comment of 04 Apr 2025, 06:32 (modified: 05 Apr 2025, 04:15)**
> >
> > Respected Authors your Fig. 1 legend reports: “Figure 1: Comparing the inference (left) and training speedups (right) (calculated using wall-clock time) of sparse training methods and the Top-1 classification accuracy (x-axis) for a ViT-Base model at 90% sparsity running ImageNet-1K.”
> >
> > To this regard I have 4 concerns:
> >
> > The Results reported in Table 11 at 90% sparsity are: dense accuracy = 78.5 > CHT = 77.66 > CHTs = 77.54 > RigL = 77.24, however your Fig. 1 reports: dense accuracy > RigL > CHT > CHTs. Please can you address this concern?
> >
> > Unfortunately at the moment the symbols indicating different methods are overlapping, generating difficulty of interpretation of the results. I would recommend reducing the size of the symbols in the revised Fig. 1 and adjusting to ensure the figure is both visually appealing and effectively communicates the intended information.
> >
> > Can you provide what setting did you use for the initializing the topology and the weight of CHT and CHTs? Did you consider hyper-parameter search of the initialization parameters and if yes what values did you use?
> >
> > As a matter of fact the goal of dynamic sparse training is to perform close or even better than dense training with a reduced number of parameters. To this regard Figure 1 reports the results at 90% sparsity which is an arbitrary value. Revising Figure 1 reporting the results at each level of sparsity mirroring the results in Table 1 would be very useful and informative. This could be achieved adding 3 small panels, reporting on the x-axis the levels of sparsity from 60% to 95% and on the y-axis in each panel a different measure such as: accuracy, inference speedup and training speedup. This could highlight already in Figure 1 that at 60% and 70% Dynamic sparse training accuracy performance is better than dense training and that DynaDiag is the only structured method to keep high its performance close to the unstructured ones that performs better than dense. I imagine DynaDiag would produce a quite stable accuracy curve that remains high in comparison to the other structured methods. It is the only structured method to present a performance stability similar to unstructured methods across different levels of sparsity. This is an impressive result that the Authors at the moment are not highlighting enough in their main article.
> >
> > **Reply to Authors rebuttal comment of 08 Apr 2025, 07:31**
> >
> > Congratulations for the job done I am raising your score to 4.
> >
> > 1. I checked the three-panel figure and it is quite evident the great performance of DynaDiag. I am happy this figure enhance the presentations of the results regarding your algorithm, I  propose to improve the figure by reporting two insets that zoon-in the results of the accuracy curves around 0.60 sparsity and around 0.95 sparsity. The same should be done for speed-up plots.
> >
> > 2. Why the input embedding layer of CHTs is not initialized using CSTI? This should increase performance of CHTs further.
> >
> > 3.  In your revised text you report that: "... propose CHT and CHTs (Zhang et al., 2024b) methods where a gradient-free (and based on the network topology) approach is used during the regrow phase, which makes their method scalable and achieves state of the art performance at high sparsity."
> > In reality looking at the results in Table 11 and in your plots the state of the art performance is obtained not only at high sparsity levels, but for each sparsity with performance at sparsity 0,6 being even larger than dense networks.
> > Can you please address this concern adjusting your sentence accordingly?
> >
> > **Reply to Authors rebuttal comment of 09 Apr 2025, 03:06**
> >
> > 1. Looking at the plots of Fig.1 that you provide, I would say that the appropriate sentence is: " ... (even outperforming the dense baseline from 60% to 80% sparsity).”
> >
> > 2. In the insets: you could add the names of the methods close to the symbols, to facilitate interpretation.
> >
> > 3. My gratitude for advancing significantly the knowledge in sparse training.
> > The score is now 5. I believe this study is exceptional and I wish you oral presentation.

---

> > > ### Author Response · Authors · 2025-04-03
> > >
> > > # Reply to reviewer’s suggestions on 08 Apr
> > >
> > > We thank the reviewer for increasing their score. We address the three queries below:
> > >
> > > 1) We have [updated the Figure 1](https://figshare.com/s/95bec1ce2911a2cc1276) with insets.
> > >
> > > 2) Our choice of using BSW initialization for CHTs is based on the text in Section 4.3 of CHTs (Zhang et al., 2024b), and from the text, we understood that CHTs can perform at par and sometimes surpass methods like CHT, which uses computationally complex methods of link prediction.
> > >
> > > 3)  Thank you for pointing this out. We propose to modify the text as follows (shortened due to space limitation):
> > >
> > > **“propose.... delivers state-of-the-art performance across all tested sparsities (even outperforming the dense baseline at 60% sparsity).**”
> > >
> > > # Reply to the reviewer's suggestions on 04 Apr
> > > We thank the reviewer for their patience and answer their concerns below:
> > >
> > > ## Current Fig 1 Results and Three Panel Fig 1
> > > Table 11 shows the correct set of results and we have fixed Figure 1 to reflect the same. Hence, the correct order of unstructured accuracy at 90% sparsity is: dense accuracy> CHT > CHTs > RigL> SET> MEST.
> > >
> > > We have also [updated current Figure 1](https://tinyurl.com/4h389z9b).
> > >
> > > ## Three panel fig1
> > > We appreciate reviewers' comments. The [new figure 1 can be accessed here](https://shorturl.at/8tQ7X). We believe that the new figure clearly shows the advantages of DynaDiag for inference and training speedup and also the superior accuracy amongst other structured DST methods.
> > >
> > > ## Init Settings and Hyperparams
> > >
> > > Based on our hyperparameter search, we use the following values:
> > >
> > > ### Topology and Weight Initialization
> > >
> > > **Topology**: For **CHT**, we use CSTI for the input embedding layer and ER for the intermediate layers (MHA) in ViT. For **CHTs**, we use a BSW ($\beta = 0.3$, obtained after a search from 0.2 to 1) based topology for all the layers.
> > >
> > > **Weight**: For both **CHT** and **CHTs**, for each sparsity level, we use the SWI method mentioned in Appendix E of CHT (Zhang et al., 2024a).
> > >
> > > ### Hyperparams
> > > Parameters for training ViTs are taken from [timm models training recipe](https://shorturl.at/hrd8t). But parameters specific to CHT and CHTs are as follows (from hyperparameter search):
> > > #### CHT
> > >
> > > $zeta = 0.3$ (Fraction removed links)
> > >
> > > $StartLR = 0.01$
> > >
> > > $EndLR = 0.0001$
> > >
> > > $UpdateInterval = 30$
> > >
> > > #### CHTs
> > >
> > > $zeta = 0.3$ (Fraction removed links)
> > >
> > > $startDelta = 0.5$
> > >
> > > $endDelta = 0.8$
> > >
> > > $K = 6$
> > >
> > > $\alpha = 1$
> > >
> > > $decayMethod = inoam$
> > >
> > > $updateInterval = 40$
> > >
> > > ###########
> > >
> > > We thank the reviewer for their support. We address the three parts of their comment below:
> > >
> > > ## Updated Figure 1
> > > We update Figure 1 with the new results, which can be [accessed here](https://figshare.com/s/3c8b904acdce99263d30).
> > >
> > > ## Updated Figure 2
> > >
> > > We update Figure 2 with the additional sparsity patterns (BSF with $\gamma$ = 2.76, BSW with $\beta$ = 0.5, $\beta$ = 1) highlighted by the reviewer. The revised figure can be [accessed here](https://figshare.com/s/9c2ecaebd44636cb3884).
> > >
> > > We will explain BSF and BSW networks in the appendix.
> > >
> > > ## Writing Edits
> > >
> > > ### Edit 1
> > >
> > > We will edit the Related Work section as follows (starting from para 4 Section 2.2. Para 1, 2 & 3 remain the same):
> > >
> > > SET (Mocanu et al., 2018) is one of the earliest DST works that introduced a prune-and-regrow strategy for DST, where during the prune phase, weights are pruned based on their magnitude and are regrown randomly. MEST (Yuan et al., 2021) regrows weights randomly and uses a combination of weight magnitude and gradient magnitude of the existing weights to prune them. RigL (Evci et al., 2020), on the other hand, prunes weights based on their magnitudes and are regrown based on the gradients of missing links (zero weights), which makes the backward pass dense and unable to take advantage of the sparsity in the network. Addressing this limitation of RigL, Zhang et al. (Zhang et al., 2024a) propose CHT and CHTs (Zhang et al., 2024b) methods where a gradient-free (and based on the network topology) approach is used during the regrow phase, which makes their method scalable and achieves state of the art performance at high sparsities.
> > >
> > > ### Edit 2
> > >
> > > We will also edit our baselines in Section 4.1 as:
> > >
> > > RigL (Evci et al., 2020), MEST (Yuan et al., 2021), SET (Mocanu et al., 2018), CHT (Zhang et al., 2024a), and CHTs (Zhang et al., 2024b) uses DST to produce unstructured sparsity, which does not yield significant speedups in training or inference.
> > >
> > > ## References
> > >
> > > [Mocanu, D. C.,et al.](https://shorturl.at/KT4G1)
> > >
> > > [Yuan, G., et al.](https://shorturl.at/wmcTB)
> > >
> > > [Evci, Utku, et al.](https://shorturl.at/O3m7g)
> > >
> > > [Zhang, Y. et al. ](https://tinyurl.com/6yswk2ru)
> > >
> > > [Zhang, Y., et al.](https://tinyurl.com/5y9mk4hp)

---

### Decision · Program_Chairs · 2025-05-01

**Decision:**

Accept (poster)

**Comment:**

This paper aims to bridge the gap between hardware, software, and the theoretical potential with respect to the efficiency of dynamic sparse training (DST) methods by proposing a new structured sparse-to-sparse DST method named DynaDiag. The paper demonstrates that DynaDiag achieves comparable or improved performance relative to dense training and structured DST methods, while providing a 1.59× training speedup on GPU. The discussion between the reviewers and the authors was productive, and the paper's scores significantly improved after the rebuttal. Therefore, the Area Chair suggests acceptance and urges the authors to include all rebuttal discussions and experimental results in the final version of the paper.